# CFAP45 deficiency causes situs abnormalities and asthenospermia by disrupting an axonemal adenine nucleotide homeostasis module

Gerard W. Dougherty et al.[#]

Axonemal dynein ATPases direct ciliary and flagellar beating via adenosine triphosphate (ATP) hydrolysis. The modulatory effect of adenosine monophosphate (AMP) and adenosine diphosphate (ADP) on flagellar beating is not fully understood. Here, we describe a deficiency of *cilia and flagella associated protein 45* (*CFAP45*) in humans and mice that presents a motile ciliopathy featuring *situs inversus totalis* and asthenospermia. CFAP45-deficient cilia and flagella show normal morphology and axonemal ultrastructure. Proteomic profiling links CFAP45 to an axonemal module including dynein ATPases and adenylate kinase as well as *CFAP52*, whose mutations cause a similar ciliopathy. CFAP45 binds AMP in vitro, consistent with structural modelling that identifies an AMP-binding interface between CFAP45 and AK8. Microtubule sliding of dyskinetic sperm from *Cfap45*$^{-/-}$ mice is rescued with the addition of either AMP or ADP with ATP, compared to ATP alone. We propose that CFAP45 supports mammalian ciliary and flagellar beating via an adenine nucleotide homeostasis module.

---

[#]A list of authors and their affiliations appears at the end of the paper.

The coordinated beating of eukaryotic cilia and flagella is regulated by axonemal heavy chain dynein ATPases, which are motor proteins that generate force along microtubules (MTs) via adenosine triphosphate (ATP) hydrolysis[1–3]. Dynein ATPases undergo cyclic conformational changes, notably the "power stroke" movement in the amino-terminal linker region, in response to binding ATP and MTs in the ATPase and MT binding domains, respectively (reviewed in ref. [4]). ATP binding and hydrolysis in the first ATPase domain (AAA1) is critical for dynein function[5,6], whereas nucleotide binding/hydrolysis in the three adjacent ATPase domains (AAA2-AAA4) likely regulate dyneins[6–9].

Since the report that adenosine diphosphate (ADP) alone can modulate the flagellar beat pattern of sperm[10], several studies have demonstrated that ADP can modulate dynein ATPase activity both in vitro and ex vivo[11–15]. The observation that ADP can rescue the inhibitory effects of high ATP concentrations on flagellar beating[16–18] suggests that both ATP and ADP cooperatively regulate dynein ATPases. It has been also shown that adenosine monophosphate (AMP) incubated with ATP can mimic the modulatory effect of ADP alone on flagellar beat pattern[19]. The mechanism underlying this effect is not fully understood but explained in part by adenylate kinase (AK), which reversibly catalyses the reaction $ATP + AMP \leftrightarrow 2\ ADP$ and maintains balanced nucleotide pools to support ciliary beating[20]. Dynein ATPases hydrolyze only one ATP per mechano-chemical cycle[21,22], raising the question as to whether ADP regulates dynein ATPases solely as a byproduct of ATP hydrolysis or distinct pools of ADP complement this function, presumably by targeting one (or more) of the four ATPase domains.

Motile ciliopathies are clinical disorders of ciliary and flagellar beating presenting as left–right asymmetry (LRA) abnormalities of the body (e.g., situs inversus totalis, situs ambiguous, cardiac malformations), dyskinetic or immotile sperm flagella (asthenospermia), and chronic upper and lower respiratory disease (reviewed in ref. [23]). Here, we investigate motile ciliopathy cases that do not fulfill the diagnostic criteria for primary ciliary dyskinesia (PCD, MIM 244400) (see also "Methods") by next-generation sequencing (NGS) and proteomic profiling of native ciliary complexes. CFAP45 loss-of-function mutations in humans as well as CRISPR/Cas9 ablation of Cfap45 in mice cause LRA abnormalities including situs inversus totalis as well as asthenospermia, due to dyskinetic beating of embryonic nodal cilia and sperm flagella, respectively. CFAP45 links dynein ATPases to an axonemal module that converges on the AK pathway. This study advances the molecular framework of mammalian ciliary and flagellar beating and presents disruption of adenine nucleotide homeostasis as a pathomechanism underlying a human motile ciliopathy.

## Results

**Loss-of-function CFAP45 mutations cause a motile ciliopathy.** We used a 772-gene "ciliaproteome" NGS panel to characterize 10 of 129 suspected motile ciliopathy cases (see "Methods"). We analyzed individual OP-28 II1 and identified compound heterozygous nonsense mutations (c.721C>T, p.Gln241* and c.907C>T, p.Arg303*, rs201144590) in cilia and flagella associated protein 45 (CFAP45; GenBank ID: NM_012337) (Fig. 1a), which we confirmed by Sanger sequencing to segregate in an autosomal recessive manner (Fig. 1b). We also analyzed 119 suspected motile ciliopathy cases by whole-exome sequencing (WES). This allowed us to prioritize CFAP45 as the likely causal gene in individual OP-985 II1, whose exome identified a homozygous frameshift mutation (c.452_464delAGAAGGAGATGGT, p. Gln151Argfs*40) that we confirmed by Sanger sequencing (Fig. 1d). In addition, a homozygous frameshift CFAP45 mutation (c.1472_1477delAGAACCinsT, p.Gln491Leufs*5) was identified

in individual TB-19 II1 that was prenatally diagnosed with LRA abnormalities including heart defect (Supplementary Fig. 1). These loss-of-function variants were either ultra-rare or absent from the gnomAD and 1000 Genomes databases (Supplementary Table 1).

All three individuals presented mild chronic upper respiratory symptoms and LRA abnormalities including situs inversus totalis (Fig. 1c and Supplementary Table 2). We analyzed respiratory ciliary beating of individual OP-28 II1 by high-speed videomicroscopy analysis (HVMA) and observed a slightly hyperkinetic ciliary beat frequency (7 Hz at 25 °C) within the average range of healthy controls (6.4 Hz at 25 °C)[24]. However, individual OP-28 II1 displayed asthenospermia with ~80% of sperm showing nonprogressive forward motility with circular or abnormal movements (Supplementary Videos 1 and 2) and abnormal flagellar waveforms including reduced curvature and angle of bending (Fig. 1e, f). In high viscosity media, the circular trajectories of OP-28 II1 sperm were corrected but the average path velocity (VAP) was ~45% slower than healthy control sperm (24 μm/s vs. 44 ± 2 μm/s) (Supplementary Fig. 2). CFAP45 has been identified in the proteomes of both human respiratory cilia and human sperm[25,26]. We detected full-length CFAP45 in both human and mouse lysates of respiratory cell and sperm samples by immunoblotting (IB) analysis, noting that smaller CFAP45 isoforms were detectable in respiratory but not sperm lysates (Fig. 1g–j).

Consistent with loss-of-function CFAP45 mutations, we verified by immunofluorescence microscopy (IFM) analysis that the panaxonemal staining of CFAP45 was undetectable in respiratory cilia from individuals OP-28 II1 and OP-985 II1 (Fig. 1l–n) as well as sperm flagella of individual OP-28 II1 (Fig. 1o–p).

**Proteomic profiling links axonemal CFAP45 to dynein ATPase components.** CFAP45 (also known as CCDC19 and NESG1) was originally cloned by mRNA differential display and proposed to function in the ciliated epithelia of the nasopharynx and trachea[27]. Because anti-human CFAP45 antibody cross-reacted with porcine respiratory cilia by IFM (Fig. 1r), we analyzed CFAP45 immunoprecipitates from isolated porcine respiratory cells (PRC) by LC/MS–MS (MS). This identified peptides corresponding to 113 protein associations over three independent experiments (Fig. 2a and Supplementary Data 1). Functional annotation of these proteins demonstrated enrichment scores (ES) for axonemal dynein complex (ES 8.91, P value 1.9E−10, Benjamini 4.5E-8, GO: 0030286) including the dynein ATPase DNAH11 and light intermediate chain dyneins DNAI1 and DNALI1, nucleobase-containing compound transport (ES 7.77, P value 1.2E-8, Benjamini 6.8E−6, GO: 0015931), and purine nucleotide binding (ES 2.59, P value 2.2E-3, Benjamini 1.2E−1, GO: 0017076) including adenylate kinase 8 (AK8).

The ciliary proteins CFAP52 and ENKUR were also identified in CFAP45 immunoprecipitates. Loss-of-function mutations in CFAP52 and ENKUR cause situs inversus totalis in humans[28,29]; ENKUR-deficient individuals show respiratory ciliary beating within normal range and do not fulfill diagnostic criteria for PCD[29]. Furthermore, the ciliary phenotype of hydrocephalus was reported in cfap52 zebrafish morphants[30] and Ak8−/− mice[31]. We confirmed that the nucleotide binding protein AK8 specifically localized to human control respiratory cilia by IFM (Fig. 2b). We also confirmed that recombinant, epitope-tagged CFAP45 reciprocally immunoprecipitated DNALI1 following their co-expression in HEK293 cells (Fig. 2c–f) and showed a binary interaction by yeast two-hybrid assay (Y2H) (Fig. 2g). Furthermore, we confirmed by IB that CFAP45 immunoprecipitated full-length DNAH11 (~520 kDa) from PRC lysates (Fig. 2h, i).

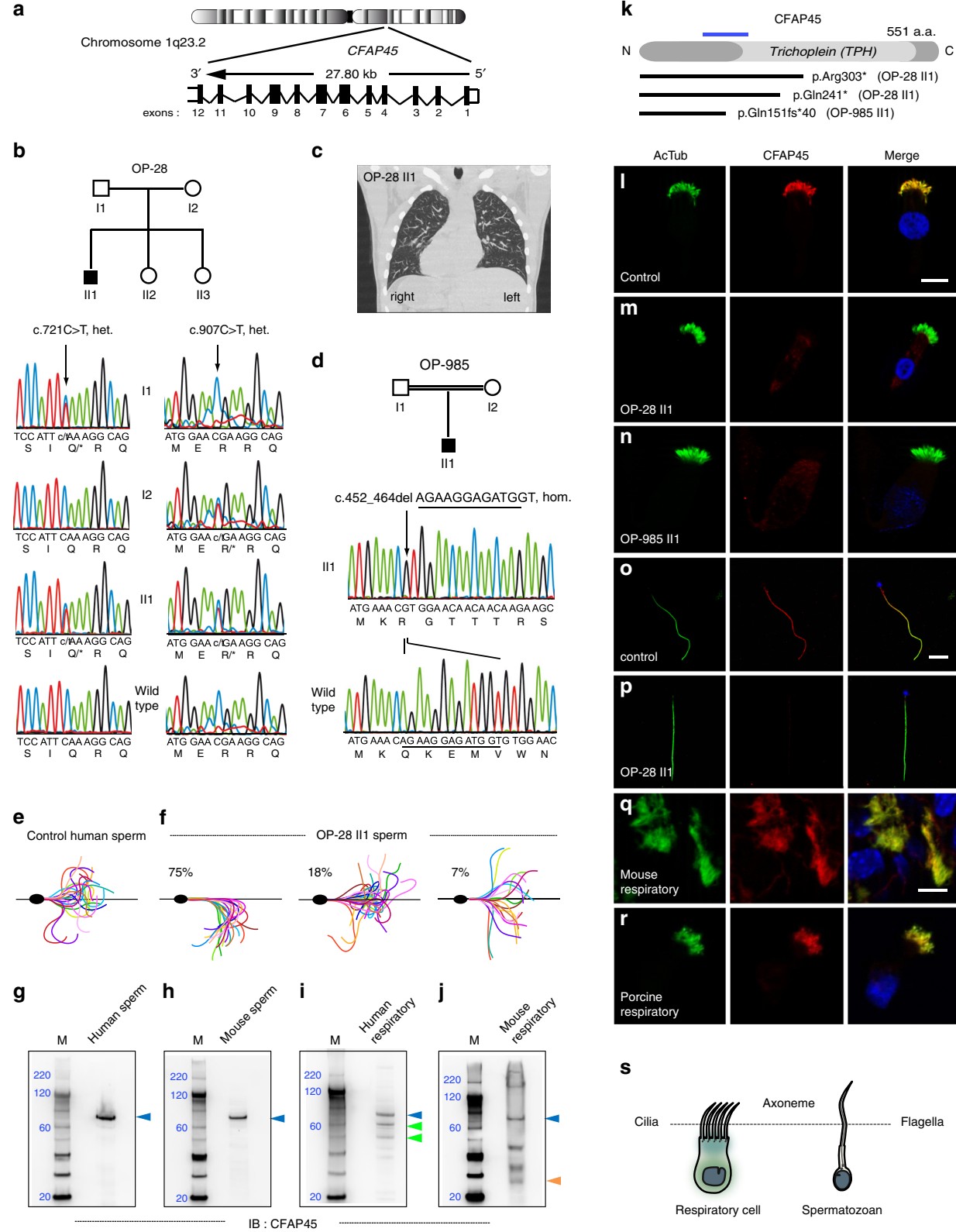

**Axonemal CFAP45 abnormalities are associated with a common cause of PCD.** Motile cilia and flagella are distinguished from primary (non-motile) cilia by complexes that mediate ciliary beating, including the outer dynein arms (ODAs), the inner dynein arms (IDAs), the radial spokes (RSs), the nexin link-dynein regulatory complex (N-DRC), and the central microtubule pair apparatus (CP) (reviewed in ref. [32]) (Fig. 3a). To determine if CFAP45 is functionally related to these complexes, we examined its localization by IFM in respiratory cilia with diverse PCD-causing mutations affecting these complexes. Notably, CFAP45

**Fig. 1 *CFAP45* mutations cause a motile ciliopathy.** Loss-of-function mutations in *CFAP45* (**a**) in individuals OP-28 II1 (**b**) showing *situs inversus totalis* (note heart positioned to right rather than left side) by computerized tomography (CT) chest scan (**c**) and OP-985 II1 (**d**). Flagellar waveforms of healthy control with normospermia (**e**) and individual OP-28 II1 with asthenospermia (**f**). **g–j** Full-length CFAP45 (blue arrowhead, approximately 66 kilodaltons) is detectable in both human and mouse sperm and respiratory lysates; CFAP45 isoforms (green and orange arrowheads) are detectable in human and mouse respiratory lysates. Marker (M) indicates relative molecular weight (blue numerals) in kilodaltons. **g–j** *n* = 1 blot for each tissue over two experiments. **k** Overview of CFAP45 protein, which has a central Trichoplein domain, indicating protein change of *CFAP45* mutations; blue line indicates relative epitope position of anti-CFAP45 antibody clone 3618 (amino acids 135–217). In contrast to control (**l**), panaxonemal CFAP45 localization (red) is undetectable in respiratory cilia from individuals OP-28 II1 (**m**) and OP-985 II1 (**n**) by IFM. In contrast to control (**o**), panaxonemal CFAP45 localization (red) is undetectable in sperm flagella from individual OP-28 II1 (**p**) by IFM. CFAP45 (red) is detectable in mouse (**q**) and porcine (**r**) respiratory cilia by IFM. Ciliary and flagellar axonemes (green) are detected using anti-acetylated α tubulin (AcTub) antibody. Merge images include Hoechst stain (blue) to indicate nuclei. White scale bars equal 10 μm. **l–r** *n* = 18 images over three experiments. **s** Reference cartoon for axonemes of respiratory cilia and sperm flagella.

retained panaxonemal localization in respiratory cilia with isolated ODA defects (*DNAH5-*, *DNAH9-*, *DNAI2-* and *CCDC151-*mutant cilia) (Fig. 3b–d and Supplementary Fig. 4), compound ODA and IDA defects ((*DNAAF1-* and *DNAAF3-*mutant cilia) (Fig. 3e, f), IDA defects with tubular disorganization (*CCDC40-* and *CCDC39-*mutant cilia) (Fig. 3g, h), and RS defects (*RSPH4A-* and *RSPH9-*mutant cilia) (Fig. 3i, j).

In contrast, CFAP45 ciliary localization was abnormal or undetectable in respiratory cilia from several genetically distinct *DNAH11-*mutant subtypes (Fig. 3k–o and Supplementary Fig. 4). Although *DNAH11-*mutant cilia appear normal by transmission electron microscopy (TEM)[33–35], a partial reduction of ODAs in a subset of axonemal microtubule doublets (MTDs) is apparent by TEM tomography[33,36]. *DNAH11* mutations are a common cause of PCD with LRA abnormalities including *situs inversus totalis*[33–36]. Similar to CFAP45, the proximal axonemal localization of DNAH11 is still detectable in respiratory cilia with the aforementioned axonemal defects[33]. Consistent with its identification by immuno-affinity capture (Fig. 2a, h), these data suggested that CFAP45 functionally interacts with the dynein ATPase DNAH11.

**Loss-of-function *CFAP52* mutations cause a motile ciliopathy.** We searched the other 118 whole exomes from our study cohort for genetic mutations in proteins identified in the CFAP45 MS dataset (Fig. 2a). This identified homozygous loss-of-function mutations in *cilia and flagella associated protein 52* or *CFAP52* (also known as *WDR16* and *WDRPUH*) among four individuals of unrelated pedigrees. Individuals OI-81 and OI-140 harbored a homozygous deletion of *CFAP52* exon 2 that we confirmed by breakpoint PCR analysis (Supplementary Fig. 5b, d). This indel (c.70 + 1535_270 + 360del, p.His25Argfs*8) has been reported in adolescent siblings with *situs inversus totalis* from an unrelated consanguineous Palestinian family[28]. Individual OI-142 harbored a homozygous frameshift *CFAP52* mutation (c.1304delG, p.Gly435Alafs*7; rs1360832162) (Supplementary Fig. 5e), and individual OI-161 harbored a homozygous missense *CFAP52* mutation (c.811G>A, p.Gly271Arg; rs140921334) (Supplementary Fig. 5f). These variants were either ultra-rare or absent from the gnomAD or 1000 Genomes databases (Supplementary Table 1).

Similar to *CFAP45-*deficient individuals, these individuals presented mild chronic upper respiratory symptoms and *situs inversus totalis* (Supplementary Fig. 5c) but did not fulfill criteria for a PCD diagnosis (Supplementary Table 2); for example, male individual OI-81 II1 showed no evidence of bronchiectasis (Supplementary Fig. 5q) but reported infertility. We detected full-length CFAP52 in lysates of human respiratory cell and sperm samples by IB, noting a smaller CFAP52 isoform in respiratory but not sperm lysates (Supplementary Fig. 5g, h). We also detected CFAP52 in human control respiratory cilia by IFM (Supplementary Fig. 5j) and confirmed that its panaxonemal localization was

undetectable in respiratory cilia from all four affected *CFAP52* individuals (Supplementary Fig. 5k–m). Furthermore, we confirmed that recombinant CFAP52 immunoprecipitated CFAP45 from HEK293 cells in an isoform-specific manner (Supplementary Fig. 3d–f) and determined that CFAP52 was abnormal or undetectable in several genetically distinct types of *DNAH11-*mutant cilia (Supplementary Fig. 6). These data demonstrated a physical–genetic interaction between CFAP45 and CFAP52 and functional cooperation with the dynein ATPase DNAH11.

**CFAP45 and CFAP52 orthologs have a conserved motile ciliary function.** To determine if CFAP45 had a conserved ciliary function in mouse, we generated CRISPR/Cas9-based mice lacking exon 3 of *Cfap45* (see Supplementary Fig. 7 and "Methods"). Similar to *CFAP45-*deficient individuals, *Cfap45*[−/−] offspring exhibited a motile ciliopathy phenotype. Consistent with *Cfap45* expression in the mouse left–right organizer (LRO) and specific CFAP45 localization to motile cilia of the LRO (Fig. 4a-f), *Cfap45*[−/−] offspring presented LRA abnormalities including *situs inversus totalis* (Fig. 4; see also "Methods"). We visualized nodal ciliary beating of *Cfap45*[−/−] embryos by HVMA and observed that their mean rotational speed per second was significantly reduced compared to heterozygous control littermates (+/−, $5.178 \pm 0.227$; −/−, $2.426 \pm 0.115$, two-tailed $t$ test, ***$p < 0.0001$) (Fig. 4h and Supplementary Videos 3 and 4), leading to impaired fluid flow at the LRO (Fig. 4i). *Cfap45*[−/−] males also displayed asthenospermia (Supplementary Videos 5 and 6); under conditions inducing hyperactivation (see "Methods"), *Cfap45*[−/−] sperm did not achieve the high amplitude and degree of flagellar bending and curvature observed in control sperm (Fig. 4k, l). This is similar to *Enkur*[−/−] sperm, which show decreased flagellar bending and curvature[37]. Although *Cfap45*[−/−] sperm appeared normal morphologically (similar to sperm from individual OP-28 II1), their mean percentage with motility (+/−, $76.72 \pm 2.697$; −/−, $32.80 \pm 2.683$, two-tailed $t$ test, ***$p = 0.0003$) (Fig. 4o) and mean curvilinear velocity (microns per second: +/−, $275.5 \pm 17.97$; −/−, $78.41 \pm 7.104$, two-tailed $t$ test, ***$p = 0.0005$) (Fig. 4p) were significantly reduced compared to littermate control. *Cfap45*[−/−] males (but not females) were infertile. Hydrocephalus was observed in both male and female *Cfap45*[−/−] mice (Supplementary Fig. 7i).

We next tested whether *CFAP45* and *CFAP52* orthologs have a conserved ciliary function in the planarian *Schmidtea mediterranea* (Smed). Planarians use ciliated epidermal cells on their ventral surface for movement, and RNAi-based targeting of genes that regulate its ciliary motility can be assayed by tracking locomotion[38,39]. Robust depletion of Smed-*cfap45* and Smed-*cfap52* by RNAi (mean expression relative to control: *cfap45*, $0.995 \pm 0.0512$ v. $0.071 \pm 0.0125$, ***$p < 0.0001$; *cfap52*, $0.846 \pm 0.107$ v. $0.028 \pm 0.015$, **$p = 0.0017$; two-tailed t test) (Fig. 5a) significantly impaired the locomotion of planaria in viscous (0.6%

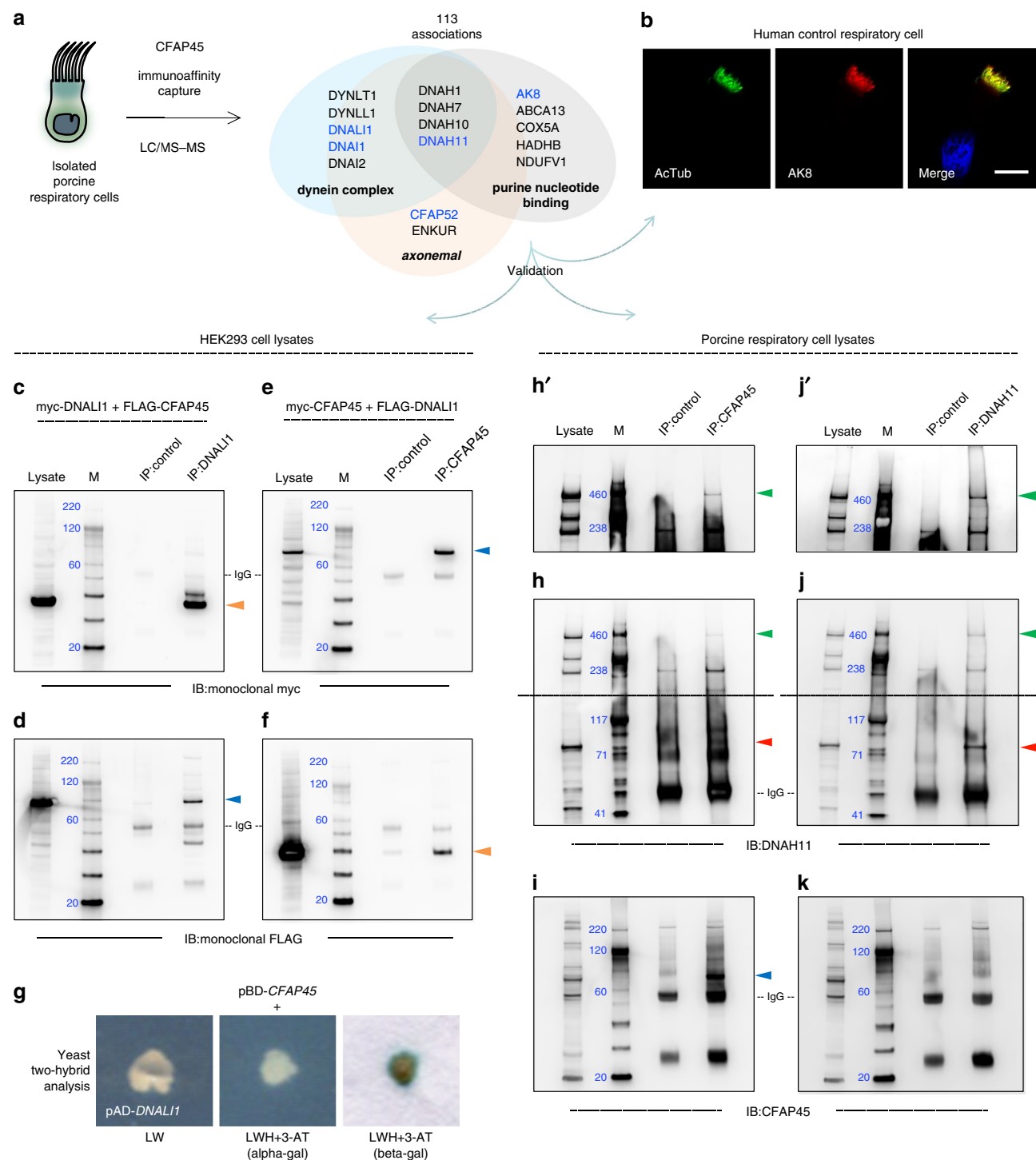

agarose) but not normal media (mean distance relative to control: *cfap45*, $1.014 \pm 0.005$ v. $0.359 \pm 0.094$, ***$p < 0.0001$; *cfap52*, $1.014 \pm 0.005$ v. $0.338 \pm 0.160$, **$p = 0.0017$; two-tailed t test) (Fig. 5b, c). TEM of these planarians showed normal ciliary ultrastructure indistinguishable from control RNAi planarians (Fig. 5d–f), consistent with TEMs of flagellar axonemes from *Cfap45*$^{-/-}$ mice (Fig. 5g, h) and respiratory cilia from individuals with loss-of-function *CFAP45* and *CFAP52* mutations (Fig. 5i–m). In all cases, TEM cross-sections of CFAP45- and CFAP52-deficient axonemes retained the typical "9 + 2" microtubular organization reminiscent of normal ciliary ultrastructure in *DNAH11*-mutant cilia. *CFAP45* and *CFAP52* are conserved in

ciliated organisms[40] and have been identified as components of flagella in *Chlamydomonas reinhardtii*[41] and motile cilia in *Trypanosoma brucei*[42] as well as upregulated transcripts of the ciliogenesis factors *FOXJ1*[43] and *MCIDAS*[44] in *Xenopus laevis*. These data supported a conserved ciliary function for *CFAP45* and *CFAP52* across eukaryotic species despite no apparent ultrastructural defects in the organization or assembly of the axoneme.

To gain insight on CFAP52 function, and because anti-CFAP52 antibody cross-reacted with porcine respiratory cilia by IFM (Supplementary Fig. 5n), we analyzed native CFAP52 immunoprecipitates of isolated PRCs by MS (Supplementary

**Fig. 2 CFAP45 interacts with axonemal dynein complex components. a** CFAP45 immunoprecipitates from isolated porcine respiratory cells (PRC) identify 113 unique associations by LC/MS–MS including *axonemal dynein complex* proteins; CFAP45 was identified in all three independent experiments. **b** AK8 (red) is detectable in human respiratory cilia by IFM. Ciliary axonemes (green) are detected using anti-acetylated α tubulin (AcTub) antibody. Merge images include Hoechst stain (blue) to indicate nuclei. White scale bar equals 10 μm. *n* = 18 images over three experiments. Polyclonal anti-DNALI1 antibody immunoprecipitates myc-tagged DNALI1 (**c**) and FLAG-tagged CFAP45 (**d**), using monoclonal anti-myc and anti-FLAG antibodies by IB, respectively. Polyclonal anti-CFAP45 antibody immunoprecipitates myc-tagged CFAP45 (**e**) and FLAG-tagged DNALI1 (**f**), using monoclonal anti-myc and anti-FLAG antibodies by IB, respectively. Recombinant DNALI1 (orange arrowhead) and CFAP45 (blue arrowhead) are ~37 and 74 kilodaltons, respectively. **c–f**, *n* = 1 blot from two experiments for each IP combination. **g** CFAP45 directly interacts with DNALI1, as indicated by growth on LWH + 3-AT media. CFAP45 also shows binary interactions to components associated with the basal body/centrosome and intraflagellar transport (see also Supplementary Fig. 3 and Supplementary Table 3). Polyclonal anti-CFAP45 antibody immunoprecipitates native DNAH11 (**h**) and CFAP45 (**i**) from PRC lysates, using polyclonal anti-DNAH11 and anti-CFAP45 antibodies by IB. For reference, polyclonal anti-DNAH11 antibody immunoprecipitates native DNAH11 (**j**) but not full-length CFAP45 (**k**) from PRC lysates. Porcine full-length CFAP45 (blue arrowhead) and DNAH11 (green arrowhead) are ~66 and 517 kilodaltons, respectively. (**h′**, **j′**) indicate longer exposure times of (**h**, **j**) cropped above gray dashed lines. A DNAH11-immunoreactive band of ~85 kDa (red arrowhead) is detectable in both CFAP45 and DNAH11 immunoprecipitates. Species-matched (rabbit) polyclonal normal IgG is used as control. Marker (M) indicates relative molecular weight (blue numerals) in kilodaltons. **h–k**, *n* = 1 blot from two experiments for each IP combination.

Fig. 8 and Supplementary Data 1). This identified peptides corresponding to 258 unique proteins over three independent experiments. Functional annotation of these proteins demonstrated enrichments for *mRNA metabolic process* (ES 17.5, *P* value 2.1E−16, Benjamini 8.6E−14, GO:0016071), *nucleic acid transport* (ES 14.79, *P* value 1.4E−15, Benjamini 4.5E−13, GO:0050657), *oxidoreductase activity* (ES 3.87, *P* value 2.3E−4, Benjamini 7.9E−3, GO:0016624) as well as for *WD40 repeat domain* (ES 5.49, *P* value 1.0E−5, Benjamini 9.2E−4, IPR001680). CFAP52 immunoprecipitates identified proteins previously associated with ciliary function, including the dynein regulatory complex protein IQCD (DRC10/CFAP84)[45] (Supplementary Fig. 8b, f–i) and interactions in common with CFAP45 including intermediate chain dynein DNAI1 and AK8. We confirmed that recombinant, epitope-tagged CFAP52 immunoprecipitated both AK8 and DNAI1 following co-expression in HEK293 cells (Supplementary Fig. 8j–l).

To further validate the association of CFAP45 and CFAP52 to ODA components including DNAH11, we performed high-resolution Airyscan imaging to determine colocalization in healthy control respiratory cilia. As shown in Fig. 6, CFAP45 and CFAP52 as well as AK8 colocalized with DNAH11 in the proximal region of respiratory cilia, comparable to colocalization of the ODA component DNAI2 with DNAH11. We also tested whether CFAP45, CFAP52, and AK8 cofractionated with dynein ATPases such as DNAH11, using dynein extracts from porcine respiratory cilia[46] fractionated across a 5–30% sucrose gradient (see "Methods"). As shown in Supplementary Fig. 9, CFAP45, CFAP52, and AK8 are detectable by IB in fractions that overlap with ODA components including DNAH11. Loss-of-function mutations in *DNAH9*, which is a paralog of *DNAH11*, cause a motile ciliopathy characterized by LRA defects and mild respiratory beating defects in the distal ciliary region[47,48]. Notably, CFAP45 retains panaxonemal localization in respiratory cilia from three unrelated and genetically distinct individuals with loss-of-function *DNAH9* mutations (Supplementary Fig. 4b–d). Because ODAs are completely absent in the distal ciliary region of *DNAH9*-mutant cilia[47], this suggests that CFAP45 specifically associates with ODAs in the proximal ciliary region via DNAH11.

**CFAP45 mediates adenine nucleotide homeostasis via AMP binding.** Curation of combined respiratory cell MS hits for CFAP45 and CFAP52 revealed a robust association to energy metabolism processes, including nucleotide binding (>50% of the combined 349 associations) and nucleic acid transport (Supplementary Data 1). We cross-referenced this with a study that identified 1157 proteins including CFAP45 and CFAP52 that are downregulated in sperm from asthenospermic males[49]. This identified a metabolic module of 108 proteins conserved in respiratory and sperm cells that featured 26 binary interactions per published interaction datasets (Supplementary Data 1). This prompted us to investigate the nucleotide-binding profiles of CFAP45 and CFAP52 using the webtool NSitePred[50]. Notably, we identified predicted binding sites to AMP and ATP for CFAP45 and CFAP52, respectively, which are well conserved among their orthologs and not retained in smaller isoforms detected in mammalian respiratory cell lysates (Fig. 5o, p; see also Fig. 1g–j and Supplementary Fig. 5g, h). Because AKs bind both AMP and ATP and participate in the reversible conversion of ATP and AMP to two ADP, we further examined the association of AK8 to both CFAP45 and CFAP52. We confirmed that recombinant full-length (p66) but not respiratory-specific, truncated (p25) CFAP45 isoforms immunoprecipitate AK8 from HEK293 cells (Supplementary Fig. 10a–d). In addition, recombinant AK8 captures both CFAP45 and CFAP52 following co-expression and tandem affinity purification (TAP) from HEK293 cells (Supplementary Fig. 10e–g). AK8 has been shown to have high affinity for AMP in vitro[51] and localize to mouse sperm flagellar axonemes[19].

We therefore tested whether CFAP45 could bind AMP, using AMP conjugated to agarose resin in four distinct orientations (Fig. 7a). CFAP45 immunoprecipitates recovered from human sperm (see "Methods") showed affinity to AMP, especially 6AH-AMP-agarose (*n* = 2) (Fig. 7b). We also utilized high salt lysis and TAP to recover recombinant CFAP45 and AK8, which preferentially showed affinity to 6AH-AMP-agarose and αAH-AMP-agarose, respectively (*n* = 2 for each) (Fig. 7d, e). We therefore hypothesized that CFAP45 supported adenine nucleotide homeostasis through an AMP-dependent process. To test this, we evaluated dynein ATPase activation via microtubule (MT) sliding of $Cfap45^{-/-}$ mouse sperm using AMP, ADP, and ATP (Fig. 7f). Administration of 1 mM ATP alone or 1 mM ADP alone did not significantly reactivate MT sliding of $Cfap45^{-/-}$ sperm compared to heterozygous control, although 1 mM ADP more robustly activated MT sliding in $Cfap45^{-/-}$ sperm compared to 1 mM ATP (Fisher's exact test, **$p$ = 0.0042) (Fig. 7g). One millimolar AMP alone showed negligible (<1%) reactivation of MT sliding in both control and $Cfap45^{-/-}$ sperm, similar to untreated (mock) control (Fig. 7g). Notably, the combination of either 4 mM AMP or 4 mM ADP with 1 mM ATP supported the partial rescue of MT sliding in $Cfap45^{-/-}$ sperm, compared to the heterozygous control (Fisher's exact test, 1 mM ATP + 4 mM AMP, *n* = 113 sperm from two males, *$p$ = 0.0432, significant; 1 mM ATP + 4 mM ADP, *n* = 93 sperm from two males, $p$ = 0.3781, not significant) (Fig. 7h). This partial rescue was not observed using 1:1 molar ratios of ATP with either

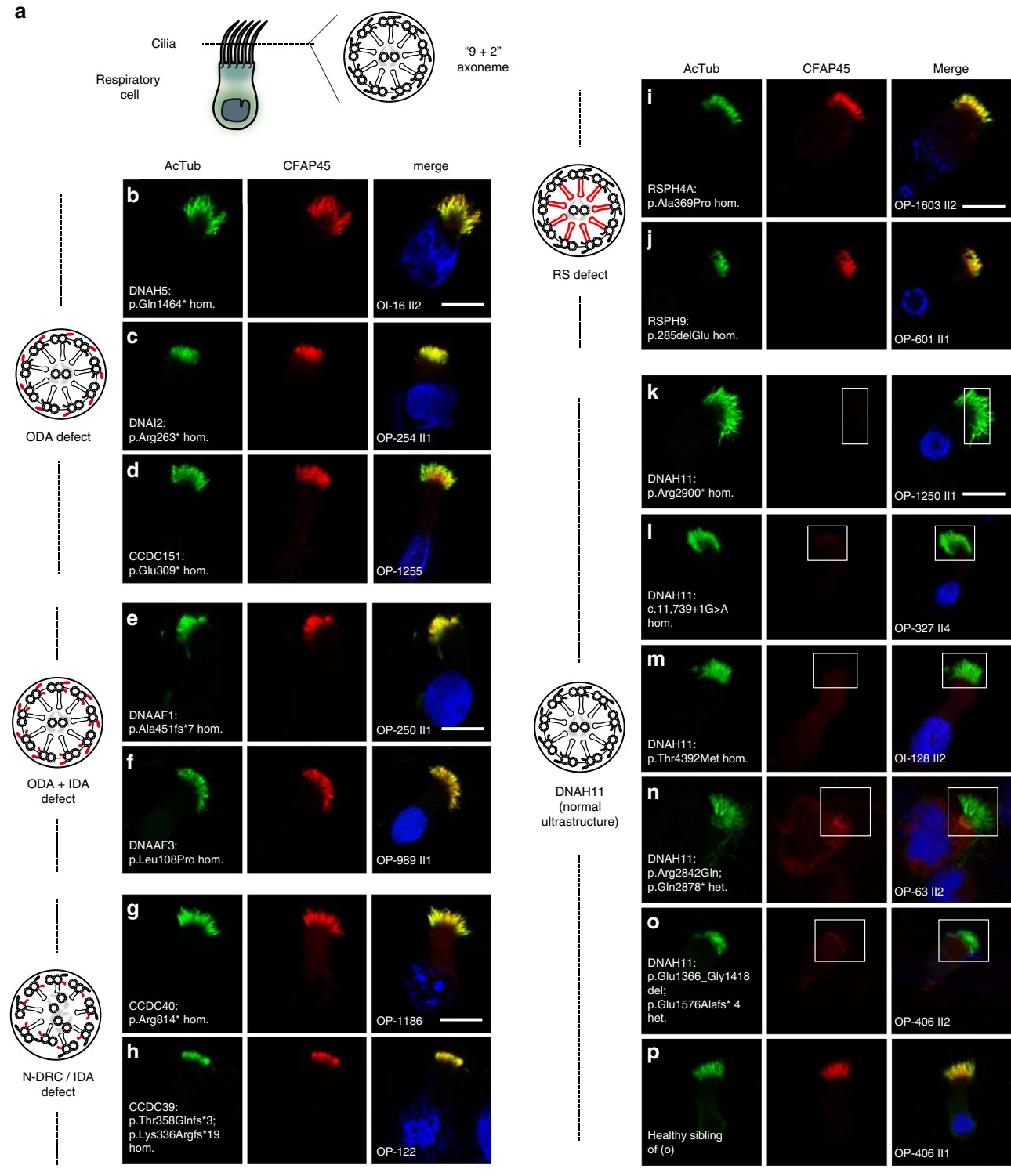

AMP or ADP (Fisher's exact test, ***$p < 0.0001$) (Fig. 7h). In our assay, nucleotide incubation spans 10 min and does not use creatine kinase to replenish ATP, allowing the AK reaction (ATP + AMP ↔ 2 ADP) to equilibrate with ATP hydrolysis by dynein ATPases. The observation that 1 mM ATP with 4 mM AMP or 4 mM ADP increased MT sliding percentage in $Cfap45^{-/-}$ sperm suggested that in part, AMP signaling may be deficient in these sperm. CFAP52 and AK8 are still detectable in sperm flagella of individual OP-28 II1 and $Cfap45^{-/-}$ males (Supplementary Fig. 10h–k), suggesting that CFAP45 deficiency does not disrupt

their axonemal localization. However, we cannot rule out that CFAP45-dependent signaling through these or other associations (e.g., IQCD) is perturbed, which may account for the partial rescue of MT sliding in response to AMP and ADP.

We next simulated AMP binding to human CFAP45 and human AK8 using isobaric–isothermal molecular dynamics (MD) simulations with FF12MC[52]. This AMBER protein forcefield has shown improved effectiveness in simulating localized disorders of folded proteins and refining comparative protein models as well as agreements between simulated and experimental folding times,

**Fig. 3 Axonemal CFAP45 localization is abnormal in certain *DNAH11*-mutant cilia by IFM. a** Cartoon of "9 + 2" axoneme in respiratory cilia. Axonemal defects that correlate with certain PCD-causing mutations are highlighted in red. Isolated ODA defects in *DNAH5*- (**b**), *DNAI2*- (**c**), and *CCDC151*-mutant (**d**) respiratory cilia do not affect axonemal CFAP45 (red) localization. Compound ODA and IDA defects in *DNAAF1*- (**e**) and *DNAAF3*-mutant (**f**) respiratory cilia do not affect axonemal CFAP45 (red) localization. N-DRC defects lacking IDAs and with tubular disorganization in *CCDC40*- (**g**) and *CCDC39*-mutant (**h**) respiratory cilia do not affect axonemal CFAP45 (red) localization. RS defects in *RSPH4A*- (**i**) and *RSPH9*-mutant (**j**) respiratory cilia do not affect axonemal CFAP45 (red) localization. **k–o** Certain *DNAH11*-mutant respiratory cilia show abnormal ciliary CFAP45 (red) localization (rectangles). **p** Healthy sibling of OP-406 II2 (**o**) shows wild type CFAP45 localization. Ciliary axonemes (green) are detected using anti-acetylated α tubulin (AcTub) antibody. Merge images include Hoechst stain (blue) to indicate nuclei. White scale bar equals 10 μm. **b–p** n = 18 images from three experiments. PCD individuals with respective mutations are indicated. *DNAH11* mutations are a common cause of PCD with LRA abnormalities[33–36] but do not perturb the axonemal localization of ODA protein DNAH5 and IDA protein DNALI1[33,35]. Per the study design, DNAH5 and DNALI1 are detectable by IFM in respiratory cilia from study cohort individuals (see "Methods"), suggesting that CFAP45 deficiency does not cause gross ODA and IDA abnormalities.

within factors of 0.69–1.75 for fast-folding proteins to fold autonomously in isobaric–isothermal MD simulations[53]. Our 18.96-microsecond MD simulations of the CFAP45·AMP·AK8 complex suggested that CFAP45 and AK8 can dimerize with favorable electrostatic and van der Waals interactions between the Trichoplein domain of CFAP45 and the second catalytic domain (KD2) of AK8 (Supplementary Fig. 11a). Consistent with AMP binding for both CFAP45 and AK8 (Fig. 7b, d, e), the microsecond MD simulations suggested that a cavity at the interface of the dimer can accommodate AMP with (1) its phosphate group tightly bound to the region surrounded by Arg479, Arg483, and Lys488 of CFAP45 and (2) its adenyl group docked at a region close to Arg367 of AK8 (Fig. 7i). Visual inspection of various AMP·CFAP45·AK8 complex conformations from the simulations suggested that the adenyl group can also dock at the region that abuts Arg357 of AK8 (Supplementary Fig. 11b), within a highly conserved AMP binding motif (Fig. 7j). Regardless of where the adenyl group docks, the phosphate group is located in the proximity of the known AMP binding site of the monomeric AK8[54]. Taken together, our 18.96-μs MD simulations suggested that CFAP45 may function as a donor of AMP to AK8. Because AMP likely does not bind directly to dynein ATPases[7], we conclude that CFAP45, as an AMP-binding protein, facilitates dynein ATPase-dependent ciliary and flagellar beating via adenine nucleotide homeostasis.

## Discussion

Here we describe *CFAP45* deficiency in man and mouse that causes a motile ciliopathy featuring LRA abnormalities (*situs inversus totalis*, heterotaxy) as well as asthenospermia. *CFAP45*-deficient individuals present mild respiratory symptoms and do not fulfill criteria for a PCD diagnosis. While *situs inversus totalis* is a relatively benign condition, heterotaxy (individual TB-19 II1) is associated with congenital heart defects as well as visceral malformations that require surgical intervention. *CFAP45* should be considered a male fertility factor, and affected males are recommended for andrological examination. In addition, we have further characterized *CFAP52* deficiency (originally reported by Ta-Shma et al.) as a motile ciliopathy that does not fulfill criteria for a PCD diagnosis.

Our data suggest a functional association between CFAP45 and dynein ATPases including ODA-associated DNAH11 along the A-tubule of axonemal MTDs. Recently, Owa et al. have characterized the Chlamydomonas orthologs to CFAP45 and CFAP52 as luminal proteins of the B-tubule, although they also identify a reduction of IDA-associated *dynein b* along the A-tubule in respective mutants[55]. Under our experimental conditions using IFM (see "Methods"), we consider the possibility that (a) the epitopes of luminal CFAP45 and CFAP52 are accessible without using non-disruptive (i.e., sarkosyl) reagents and/or (b) that distinct isoforms of mammalian CFAP45 and CFAP52 have adapted specific functions for the A- and B-tubules of axonemal MTDs. There are

notable structural and molecular differences between Chlamydomonas and mammalian axonemes, including the reduction of a three-headed ODA (α, β, and γ heavy chain genes) in Chlamydomonas to a two-headed ODA (orthologs of β and γ heavy chain genes) in mammals[56]. Furthermore, whereas the α heavy chain gene has no clear ortholog in higher eukaryotes, the β heavy chain gene has duplicated not once but twice[57] to reflect tissue-and axoneme-specific functions for DNAH11 and DNAH9 in the proximal and distal ciliary regions of mammalian respiratory cilia[33,47,48] as well as DNAH17 in mammalian sperm flagella (Supplementary Fig. 12). A recent study has identified *DNAH17* mutations in males with asthenospermia and characterized DNAH17 as the functional paralog of ODAs in human spermatozoa[58]. Interestingly, we also identified CFAP45 associations to IDA-associated DNAH7 and DNAH10 in porcine respiratory lysates (Supplementary Data 1 and Supplementary Fig. 13). Future studies such as proteomic and cryo-EM analysis of CFAP45- and CFAP52-deficient mammalian cilia and sperm may be helpful in clarifying their functional roles at the A- and B-tubule of axonemal MTDs.

Our data also suggest that CFAP45 mediates adenine nucleotide homeostasis, in part through AMP binding and its association with AK8. Additional studies are required to determine if AMP (or other nucleotides) affects the function of CFAP45 independent of its association with AK8, as well as influencing kinetic parameters of the AK reaction. The partially restorative effects of AMP and ADP on MT sliding of CFAP45-deficient sperm as well as modeling of the CFAP45•AMP•AK8 interface lead us to speculate that CFAP45 may act as an AMP donor to AK8 and promote the forward reaction toward ADP production. In turn, this potentially explains a mechanism for dynein ATPase regulation, albeit indirectly.

Substantial evidence supports the role of ADP in the regulation of dynein ATPases, including its effect on ATP hydrolysis, processivity, and affinity for MTs. In a key study, Kinoshita et al. demonstrated that ADP could promote MT sliding of ciliary axonemes in vitro as well as rescue the inhibition of this sliding under high ATP concentration, which requires an intact AK pathway[17]. Mocz and Gibbons demonstrated by partition analysis that isolated sea urchin dynein ATPases could appreciably bind both ATP and ADP (at levels about 75% that of ATP) but not AMP[7]. Studies with isolated *Chlamydomonas* and *Tetrahymena* dynein *a* demonstrated a role for ADP in providing force to translocate MTs in vitro[11,12], and studies with sea urchin sperm demonstrated a critical role for ADP in bending flagella by using very low ATP concentrations[59] as well as by enzymatically blocking ATP regeneration[60]. These studies led to the prediction that one or more of the four ATPase domains contain a high affinity ADP binding site[11,13,60]. This is consistent with our analysis of the four ATPase domains among all human axonemal and cytoplasmic dynein ATPases using NSitePred, which identifies higher probability for binding ADP than ATP in certain ATPase domains (Supplementary Fig. 14).

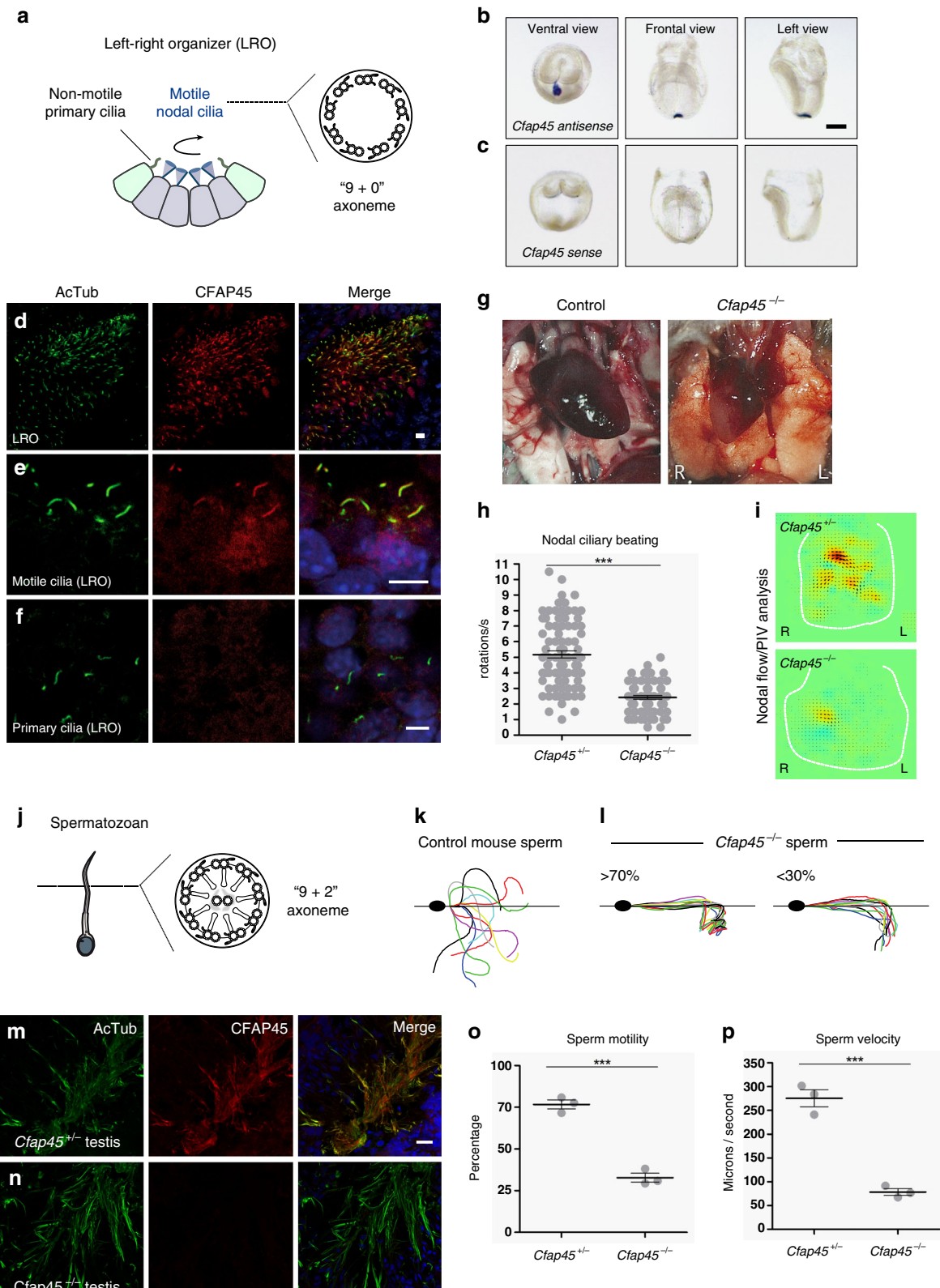

Although ATP hydrolysis has been reported in the AAA3 and AAA4 domains, the AAA1 domain is the primary site of hydrolysis and is critical for dynein ATPase-mediated motility[5,6,8]. Because dynein ATPases only hydrolyze one ATP per cycle[21], it has been speculated that the AAA3 and AAA4 domains serve regulatory functions and hydrolyze ATP less frequently than the AAA1 domain (reviewed in ref. [4]).

Saccharomyces dynein ATPase constructs with ATP-hydrolysis mutations in the AAA3 and AAA4 domains showed an approximately twofold reduction in force production, demonstrating that trapping these domains in an ATP-bound state could allosterically modulate MT binding affinity and force production by decreasing ATP hydrolysis at the AAA1 domain[8].

**Fig. 4 *Cfap45*$^{-/-}$ mice replicate a motile ciliopathy phenotype. a–f** Motile cilia of the LRO have a "9 + 0" axoneme (**a**). *Cfap45* is detectable in the mouse embryonic LRO by in situ hybridization (antisense probe) (**b**); control probe (sense) shows no hybridization (**c**). Black scale bar equals 100 µm. **d–f** CFAP45 (red) localizes to motile LRO cilia (**d**, **e**) but not non-motile (primary) cilia (**f**) of the LRO by IFM. **g–i** Compared to wild-type control, *Cfap45*$^{-/-}$ mice show LRA defects including *situs inversus totalis* with heart malpositioned on the right side (**g**). **h** Rotational speed of motile LRO cilia is significantly reduced in *Cfap45*$^{-/-}$ embryos ($n = 74$ cilia from four embryos) compared to control ($n = 87$ cilia from 4 embryos); data are mean ± SEM, significance assessed by two-tailed t test (***$p < 0.0001$). See also Supplementary Videos 3 and 4. **i** Particle image velocimetry (PIV) analysis shows significantly reduced fluid flow at the LRO of *Cfap45*$^{-/-}$ embryos compared to control. Yellow and red color indicate leftward flow, blue indicates rightward flow. **j–p** Similar to respiratory cilia, sperm flagella have a "9 + 2" axoneme (**j**). In contrast to control (**k**), *Cfap45*$^{-/-}$ sperm flagellar waveforms show reduced curvature and amplitude under conditions inducing hyperactivation (**l**). **m**, **n** CFAP45 (red) is detectable by IFM in testis cryosections of heterozygous littermates (**m**) but not *Cfap45*$^{-/-}$ males (**n**). **o** Compared to control, motility of *Cfap45*$^{-/-}$ sperm is significantly reduced ($n = 195$ sperm over three independent experiments for each; data are mean ± SEM; significance assessed by two-tailed t test, ***$p = 0.0003$). **p** Compared to *Cfap45*$^{+/-}$ littermates, *Cfap45*$^{-/-}$ sperm show significantly reduced curvilinear velocity ($n = 99$ or 70 sperm over three independent experiments for *Cfap45*$^{+/-}$ and *Cfap45*$^{-/-}$, respectively; data are mean ± SEM; significance assessed by two-tailed t test, ***$p = 0.0005$). See also Supplementary Videos 5 and 6. Ciliary and flagellar axonemes (green) are detected using anti-acetylated α tubulin (AcTub) antibody. Merge images include Hoechst stain (blue) to indicate nuclei. White scale bars equal 10 µm. $n = 6$ images from 2 experiments (**b**, **c**, **m**, **n**); $n = 4$ images from one experiment in **d**, **f**.

Seminal studies by the Lindemann laboratory have demonstrated ex vivo that ADP is critical in coordinating the flagellar beat cycle by providing force to overcome the "switch point" or transition from forward to reverse flagellar bending. ADP induces a rhythmic beating to flagella when supplemented with ATP[10] and can increase the curvature and bending angle of the beat cycle using higher concentrations of ADP[15,18,61]. Their observations that ADP markedly increases the transverse tension (t-force) on the flagellar axoneme and results in a greater number of dynein ATPases tightly bound to MTs supported initial findings by Inoue et al.[14] and largely bore out predictions of the "geometric-clutch" hypothesis[62].

We consider one hypothetical model whereby *CFAP45* deficiency disrupts ADP homeostasis and indirectly perturbs the function of ADP-sensitive regulatory domains of certain dynein ATPases (Supplementary Fig. 15). In this scenario, ATP could occupy these regulatory domains by default, partially mimicking the weaker force production of ATP hydrolysis mutants AAA3 and AAA4[8]. This is compatible with slower rotational beating in *Cfap45*$^{-/-}$ LRO cilia (Fig. 4h, i and Supplementary Videos 3 and 4) and abnormal flagellar waveforms in CFAP45-deficient sperm (Figs. 1f and 4l), which do not show the characteristic curvature and bending angle that is induced with ADP[15]. This also helps explain why 1 mM ATP alone does not significantly rescue MT sliding of *Cfap45*$^{-/-}$ sperm (Fig. 7g). We detect isoforms of CFAP45 and CFAP52 in respiratory but not sperm lysates (Fig. 1i, j and Supplementary Fig. 5g, h), suggesting tissue-specific functions via alternative splicing may partially explain the "discordant" ciliary phenotype observed in CFAP45- and CFAP52-deficient individuals. This is supported by the demonstration that a smaller respiratory CFAP45 isoform lacking its Trichoplein domain (harboring the conserved AMP binding motif) does not associate with AK8 (Supplementary Fig. 10c, d). Overall, these findings provide insights into the modulatory effects of AMP and ADP on flagellar beating and further classify the clinical and genetic spectrum underlying human motile ciliopathies.

## Methods

**Participation of ciliopathy individuals in the PCD study.** This study complies with guidelines provided by Strengthening the Reporting of Observational Studies in Epidemiology (STROBE) for cohort studies. Individuals suspected to have a motile ciliopathy participated in PCD-study protocols that were approved by the Institutional Ethics Review Board from the University Hospital of Muenster (Germany) and collaborating institutions. The PCD protocol includes clinical examination for respiratory symptoms, X-ray or CT scan for assessment of situs abnormalities, measurement of nasal nitric oxide (nNO), blood withdrawal for genetic analysis, and nasal brush biopsy to analyze cellular, molecular, and ultrastructural features of ciliated respiratory cells. Inclusion criteria were chronic upper respiratory symptoms (e.g., rhinosinusitis, otitis media) and LRA abnormalities

(e.g., *situs inversus totalis*) with or without asthenospermia/male infertility. The diagnosis of PCD is supported by evidence of destructive lower airway disease (bronchiectasis), hallmark features of dyskinetic or immotile respiratory cilia by HVMA, low nNO levels, and structural defects of respiratory cilia by conventional transmission electron microscopy (TEM) analysis[63,64]. Individuals without symptoms of chronic destructive airway disease as well as HVMA and nNO levels within normal range do not fulfill diagnostic criteria for PCD. Individuals provided signed and informed consent to participate in this study and use their anonymized data to publish intermediate results.

**Study design.** IFM analysis of isolated respiratory cells can identify molecular defects that support a PCD diagnosis and prioritize genetic characterization of motile ciliopathies including PCD. Two of the most common structural defects in PCD-affected cilia, absent ODAs (>50% of PCD cases, our cohort analysis) and absent IDAs with microtubular disorganization (>10% of PCD cases), can be prioritized based on a negative/abnormal IFM pattern using antibodies to the dynein axonemal heavy chain protein DNAH5[65] and the dynein axonemal light intermediate chain protein DNALI1[66], respectively. Furthermore, an abnormal/negative IFM pattern for both DNAH5 and DNALI1 could indicate a compound ODA and IDA defect (<10% of PCD cases) resulting from mutations in dynein-associated assembly factors (DNAAFs)[67].

We analyzed respiratory cells from individuals that were excluded for a PCD diagnosis by IFM and determined that DNAH5 and DNALI1 were detectable in respiratory cilia, suggesting normal ciliary structure. These individuals ($n = 129$) represented the motile ciliopathy (non PCD) study cohort and were prioritized for mutational analysis using a NGS approach. Selected cases were analyzed using either a customized "ciliaproteome" 772-gene panel[33,68] or by WES. This study cohort would likely exclude mutations representing the most common causes of PCD, including: isolated ODA defects (e.g., *DNAH5*, MIM 603335), IDA defects with microtubular disorganization (e.g., *CCDC39*, MIM 613798), and combined ODA/IDA defects (e.g., *DNAAF2*, MIM 612517) as well as mutations affecting RS, N-DRC and CP structures, which have not been associated with LRA defects. In a separate cohort and in collaboration with Hadassah-Hebrew University Medical Center (Jerusalem, Israel), individuals diagnosed with LRA abnormalities and suspected for consanguinity were prioritized for genetic analysis by WES. Individual TB-19 II1 was prenatally diagnosed with heterotaxy syndrome including heart defect that subsequently required corrective surgery. Parental consent was provided to analyze DNA from individual TB-19 II1 by WES as part of ongoing studies at Hadassah-Hebrew University Medical Center to determine the genetic basis of LRA abnormalities.

Targeted mutational analysis of 772 genes prioritized as likely functional ciliary components[68] to analyze ciliopathy cases by both copy-number variant analysis (array comparative genomic hybridridization) and targeted exon sequencing (NGS) were described[33]. We used this "ciliaproteome" NGS panel to analyze 10 of 129 suspected motile ciliopathy cases. Briefly, an Illumina HiSeq platform was used for targeted liquid capture and resequencing. We conducted pre-capture pooling of samples including DNA from individual OP-28 II1 (23 samples per pool) using a custom NimbleGen targeted liquid capture library targeting the coding regions and splice junctions of *CFAP45* (*CCDC19*). We generated paired-end 100 base pair reads on a HiSeq2000 instrument (Illumina), resulting in a mean of 110× coverage with 92% of bases achieving >20× coverage in 99% of targeted regions captured. Sanger sequencing confirmed that these mutations segregated in an autosomal recessive manner (Fig. 1).

WES was performed as described[33]. We searched 119 exomes from this study cohort for *CFAP45* mutations, using these 772 genes as a first-pass filter for pathogenic variants identified by WES. This allowed us to prioritize *CFAP45* as the likely causal gene in individual OP-985 II1 (Fig. 1), whose exome identified a homozygous frameshift mutation (c.452_464delAGAAGGAGATGGT,

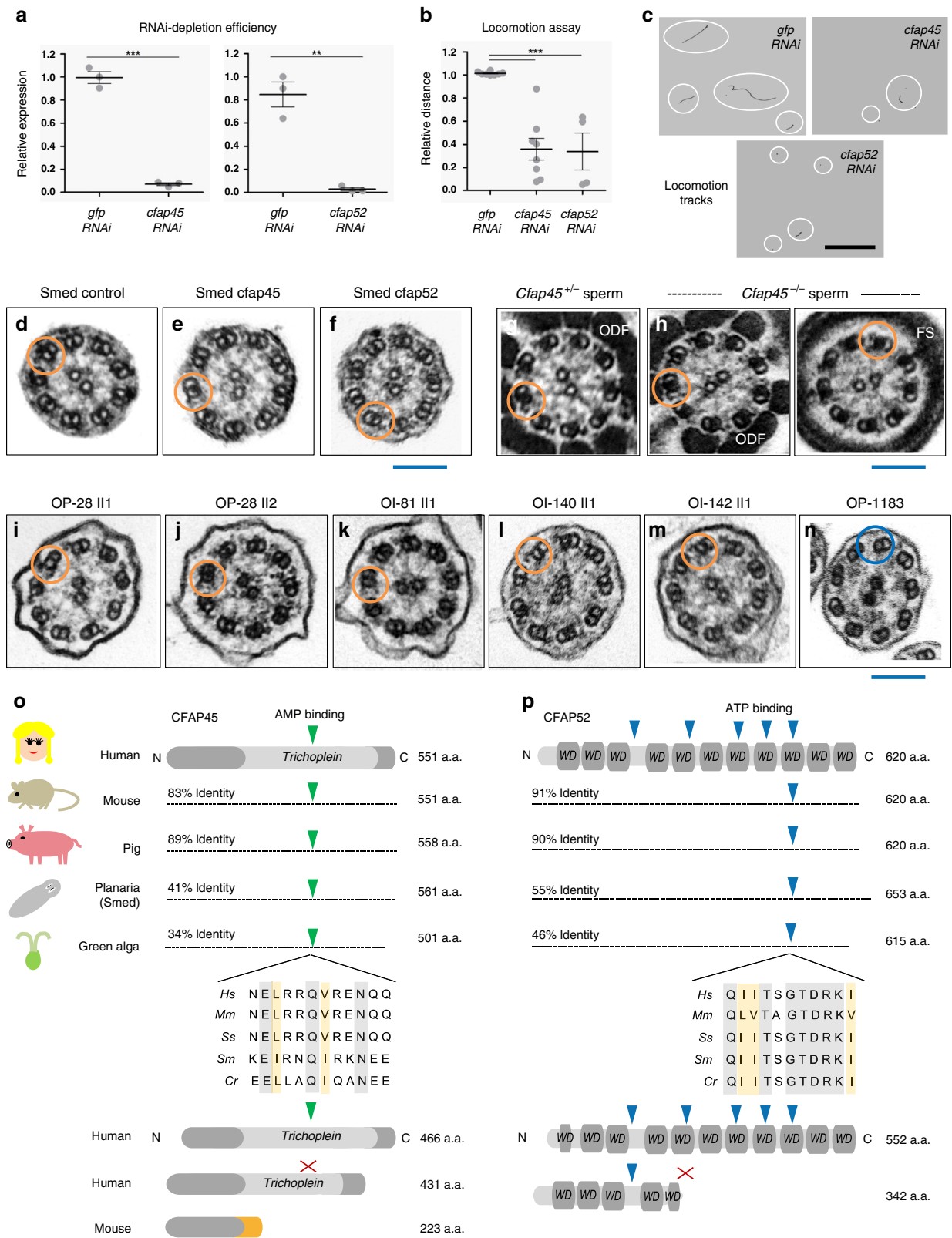

p.Gln151Argfs*40) that was confirmed by Sanger sequencing (Fig. 1). Due to the association between CFAP45 and CFAP52 (Fig. 2 and Supplementary Fig. 3), we searched the other 118 exomes from this study cohort for *CFAP52* mutations. Sanger sequencing confirmed that these mutations segregated in an autosomal recessive manner (Supplementary Fig. 5).

Briefly, WES of genomic DNA from ciliopathy individuals was performed at the Wellcome Trust Sanger Institute. Fragments of genomic DNA averaging 150 base pairs were used to create a DNA library using established Illumina paired-end

protocols. Adapter-ligated libraries were amplified and indexed by PCR, and a portion of each library was used to create an equimolar pool comprising eight indexed libraries. The exomes were produced using Illumina HiSeq and SureSelect Human All Exon v5 (Agilent) SeqCap pulldown technology. Two callers were used and their results merged for each sample separately: an all-sites BCF was created with samtools mpileup (samtools version 0.1.18) and then variants (SNPs and indels) were called by bcftools; also variant sites (SNPs and Indels) were called using the GATK Unified Genotyper (GATK version 1.3-21). The variants called by

**Fig. 5 CFAP45 and CFAP52 orthologs have a conserved ciliary function. a** RNAi efficiently depletes *cfap45* and *cfap52* in *Schmidtea mediterranea* (Smed) ($n = 3$ animals over three independent experiments for *cfap45*, *cfap52*, and *gfp* control); data are the means ± SEM; significance is assessed by two-tailed *t*-test (***$p < 0.0001$; **$p = 0.0017$). **b** In contrast to *gfp*- ($n = 62$ animals over eight independent experiments), the locomotion of *cfap45*- ($n = 65$ animals over eight independent experiments) and *cfap52*-RNAi planarians ($n = 33$ animals over four independent experiments) is significantly impaired in 0.6% low-melting agarose; data are the means ± SEM; significance is assessed by two-tailed *t*-test (***$p < 0.0001$). **c** Representative locomotion tracks (white circles) of *gfp, cfap45, and cfap52*-RNAi planarians; black scale bar equals 33 mm. **d**–**n** CFAP45- and CFAP52-deficient axonemes across eukaryota show normal ciliary ultrastructure. Similar to *gfp* control (**d**), TEM shows normal ciliary ultrastructure in *cfap45* (**e**) and *cfap52* (**f**) RNAi planarians. $n = 8$ images from two experiments for *gfp, cfap45*, and *cfap52*. Similar to heterozygous littermates (**g**), *Cfap45*$^{-/-}$ flagellar axonemes show normal ultrastructure in the midpiece (ODF outer dense fiber) and principal piece (FS fibrous sheath) regions, respectively (**h**). $n = 8$ images from two experiments for *Cfap45*$^{+/-}$ and *Cfap45*$^{-/-}$ sperm. Axonemal ultrastructure of respiratory cilia from CFAP45-deficient individual OP-28 II1 (**i**) is indistinguishable from healthy sibling OP-28 II2 (**j**) as well as CFAP52-deficient individuals OI-81 (**k**), OI-140 (**l**) and OI-142 (**m**). For reference, PCD individual OP-1183 (LRRC6; p.Gln188*hom.) shows a typical ciliary ultrastructure lacking both ODAs and IDAs (**n**). $n = 6$ images from 1 experiment for (**i**–**n**). Orange rings indicate detectable ODA and IDA pairs; blue ring indicates absent ODA and IDA pairs. Blue scale bars equal 100 nm. Potential AMP (green arrowhead) and ATP-binding (blue arrowhead) sites are conserved in CFAP45 (**o**) and CFAP52 (**p**) orthologs, respectively, in an isoform-specific manner. See also Fig. 1g–j and Supplementary Fig. 5g, h. Protein alignments highlight identity (gray shade) and similarity (orange shade). Red "X" indicates no predicted binding site.

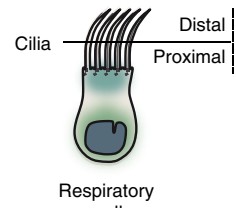

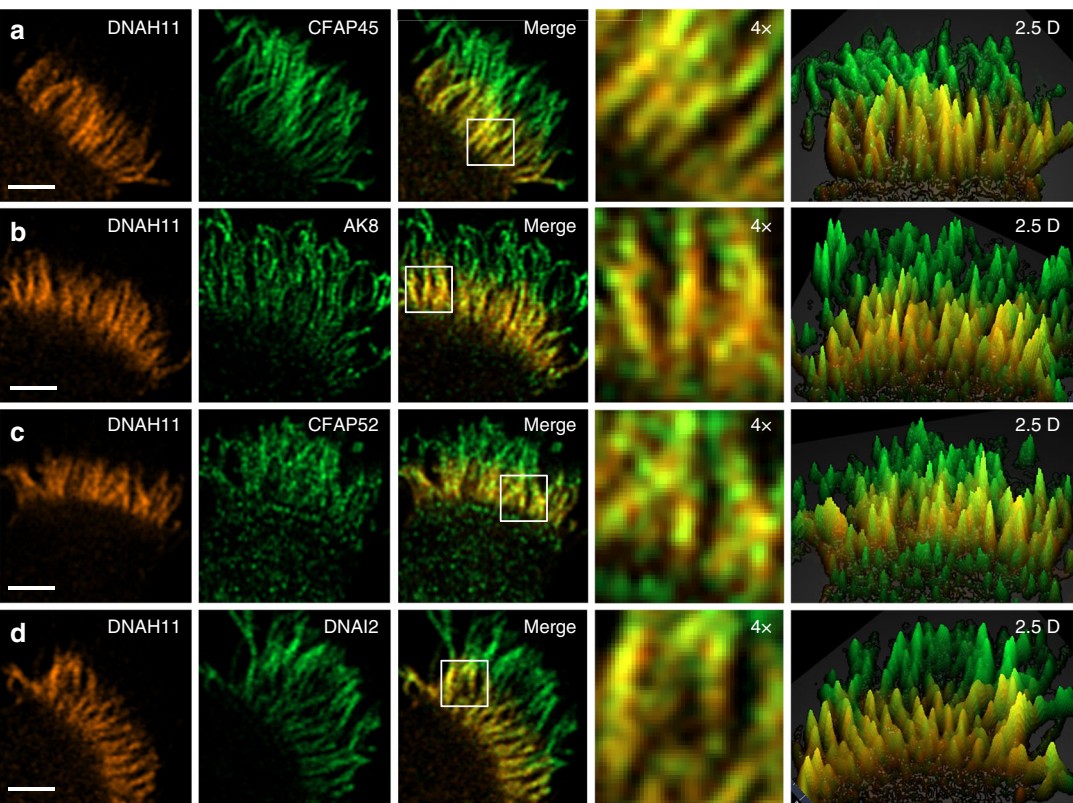

**Fig. 6 CFAP45 and CFAP52 as well as AK8 colocalize with DNAH11 in the proximal region of respiratory cilia.** High-resolution Airyscan imaging of healthy control respiratory cilia shows that DNAH11 colocalizes with CFAP45 (**a**), AK8 (**b**), and CFAP52 (**c**) as well as the ODA component DNAI2 (**d**) in the proximal ciliary region. Merge panels show colocalization (yellow) in the proximal ciliary region. DNAH11 (red) is detected using monoclonal anti-DNAH11; CFAP45, CFAP52, AK8, and DNAI2 (green) are detected using respective polyclonal antibodies at a final concentration of 0.25 μg/ml. Average colocalization coefficients with reference to DNAH11 (red, Alexa Fluor 546) are as follows: CFAP45, $n = 15$ from two experiments (0.564, 0.663 weighted), CFAP52, $n = 13$ from two experiments (0.878, 0.913 weighted), AK8, $n = 13$ from two experiments (0.799, 0.835), DNAI2, $n = 15$ from two experiments (0.760, 0.820 weighted). White scale bar equals 2 μm. 4× indicates four times magnification of the white box in merge panel, which represents two square microns. Two and a half dimensional (2.5D) representations show intensity values as projections in the *Z*-axis (grid distance 1%).

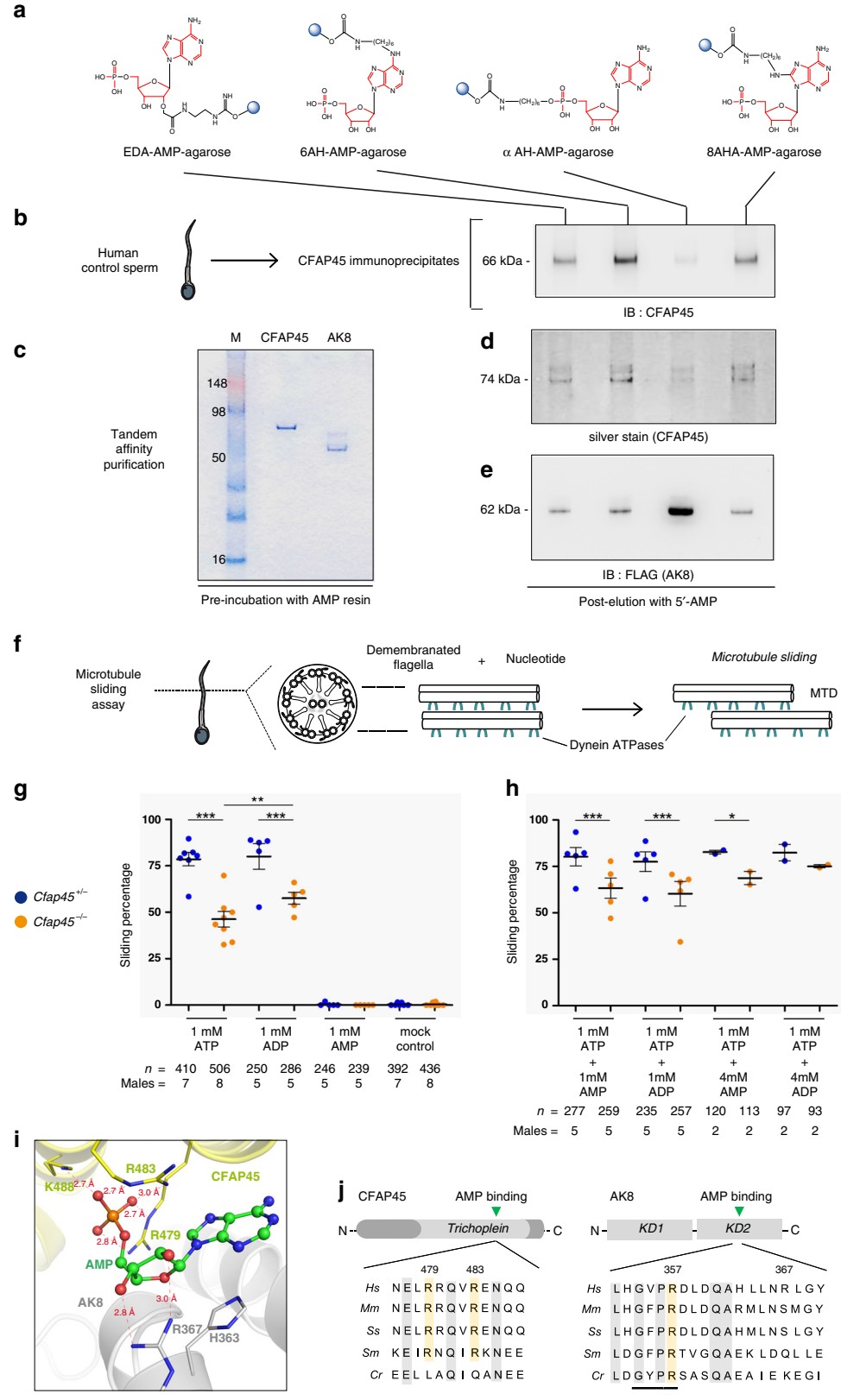

each of the callers were (soft) filtered separately and annotated using vcf-annotate (VCFtools). The VCF files of OP-985 II1, OI-81, OI-140, OI-142, and OI-161 were analyzed using Varbank software (Cologne Center for Genomics). See also Supplementary Table 1.

Variant analysis was performed as described[33]. Briefly, potential ciliopathy-causing variants identified by NGS were filtered against the 772 ciliaproteome gene list (available upon request) using Varbank software. Only variants with an average depth of coverage exceeding 30× were considered. Potential pathogenic variants

were prioritized if (a) mode of inheritance was consistent with compound heterozygosity or homozygosity, (b) the minor allele frequency was below 1% (0.01) per the 1000 Genomes and gnomAD databases, and (c) the protein change was classified as deleterious, probably damaging, or possibly damaging according to PROVEAN and Polyphen-2 analysis (neutral and benign classifications were not considered further). A homozygous missense variant (rs80010329) in *SPEF2* was also identified in CFAP52-deficient individual OI-140 II1 (Supplementary Fig. 5d). *Spef2*$^{-/-}$ (bgh) mice do not manifest LRA abnormalities[69]; *SPEF2* mutations are

**Fig. 7 CFAP45 mediates adenine nucleotide homeostasis via AMP binding. a** AMP-binding screen using AMP conjugated to agarose (blue circles) in four orientations (see also "Methods"). **b** CFAP45 immunoprecipitates from human sperm show affinity to AMP including 6AH-AMP-agarose following elution and IB with anti-CFAP45 antibody. **c** Coomasie-stained gel shows recombinant CFAP45- and AK8-enriched lysates recovered by TAP. **d** NTAP-CFAP45 shows affinity to AMP including 6AH-AMP-agarose by silver stain. **e** NTAP-AK8 shows affinity to αAH-AMP-agarose following elution and IB with monoclonal anti-FLAG antibody. **f** MT-sliding assay to reconstitute dynein ATPase activity. **g** 1 mM ATP or 1 mM ADP alone does not significantly reactivate sliding percentage of $Cfap45^{-/-}$ sperm compared to control (***$p < 0.0001$); however, 1 mM ADP significantly increases sliding percentage of $Cfap45^{-/-}$ sperm compared to 1 mM ATP (comparison of $+/-$, $p = 0.3702$, not significant; $-/-$, **$p = 0.0042$; data are means ± SEM; significance assessed by Fisher's exact test); 1 mM AMP alone shows negligible (<1%) reactivation, similar to untreated control. **h** Combination of either 4 mM AMP ($n = 113$ sperm from 2 experiments, *$p = 0.0432$) or 4 mM ADP ($n = 93$ sperm over 2 experiments, $p = 0.3781$) with 1 mM ATP increases sliding percentage of $Cfap45^{-/-}$ sperm, compared to heterozygous control (data are the means ± SEM; significance supported by Fisher's exact test); equimolar ratios of 1 mM AMP or 1 mM ADP with 1 mM ATP does not significantly reactivate $Cfap45^{-/-}$ sperm sliding percentage compared to control (data are means ± SEM; significance assessed by Fisher's exact test; ***$p < 0.0001$). **i** Close-up view of AMP bound at the interface of CFAP45•AK8 complex. Carbon atoms in AMP, CFAP45, and AK8 are colored in green, yellow, and gray, respectively; nitrogen, oxygen, and phosphorus atoms are colored in blue, red, and orange, respectively. Distances shown by thin-dash lines are in Angstroms (Å). **j** Sequence alignments highlighting highly conserved CFAP45 residues Arg479 and Arg483- and AMP-binding site of AK8 (residues 354-357, black bar). Conserved residues highlighted in gray; arginine residues highlighted in orange. See also Supplementary Fig. 11 and "Methods".

not consistent with the clinical presentation of *situs inversus totalis* observed in individual OI-140 II1.

Sanger sequencing was performed as described[33]. The reference sequences are as follows: CFAP45, ENST00000368099 (Ensembl)/NM_012337 (NCBI); CFAP52, ENST00000396219.7 (Ensembl)/NM_001080556 (NCBI). Forward (F) and reverse (R) primers used to bidirectionally analyze genomic fragments of coding exons for *CFAP45* and *CFAP52* are listed below:

*CFAP45* exon 1-F: 5′-GGCTAGTGGTTGCCAAGGTA-3′
*CFAP45* exon 1-R: 5′-CCTGACCCCTAGACCCAACT-3′
*CFAP45* exon 2-F: 5′-CCCCTATGGTTCAGAGTTTACACAG-3′
*CFAP45* exon 2-R: 5′-GACATGAGAGATTCTAGGGTGGAAG-3′
*CFAP45* exon 3-F: 5′-GGACTGACCAGCAGGAAGTG-3′
*CFAP45* exon 3-R: 5′-TGGGTTTACCCATCTCCCTA-3′
*CFAP45* exon 4-F: 5′-ATGGGAAGAGGGGAAGAGAG-3′
*CFAP45* exon 4-R: 5′-AGGGTCCCCTGATGAGAGTT-3′
*CFAP45* exon 5-F: 5′-GGCTCTGACATCATCACTGG-3′
*CFAP45* exon 5-R: 5′-CATGGTCACACAGCGAACA-3′
*CFAP45* exon 6-F: 5′-CTCCAGCATGCTCAGCAGTA-3′
*CFAP45* exon 6-R: 5′-TCTCTTCCCTCTTTGCTAGGC-3′
*CFAP45* exon 7-F: 5′-TCTTGGGGGTTGTTTAGGTG-3′
*CFAP45* exon 7-R: 5′-GTGCCTGGCAGAAAATCAGT-3′
*CFAP45* exon 8-F: 5′-CCCTCCTCTGTCTGCATCTC-3′
*CFAP45* exon 8-R: 5′-AGTTCTCAGGCCCTCCTCTC-3′
*CFAP45* exon 9-F: 5′-GCACCTGGCCTTATTTTGTC-3′
*CFAP45* exon 9-R: 5′-CATCCTTTCCTCCGCATCT-3′
*CFAP45* exon 10-F: 5′-GCCAGAGAAATTCATGATGCT-3′
*CFAP45* exon 10-R: 5′-ACCCAGACAAGATCGACGAG-3′
*CFAP45* exon 11-F: 5′-TCTGAGGCATGTCTACTGAAGG-3′
*CFAP45* exon 11-R: 5′-AGAAGAGCCTCTCCCTGAGC-3′
*CFAP45* exon 12-F: 5′-TCTGTAAACCCCACCCAGAG-3′
*CFAP45* exon 12-R: 5′-TCCCAGGGAATCACAAAAG-3′
*CFAP52* exon 1-F: 5′-CATCCCCTAAATTTCCAGAGAA-3′
*CFAP52* exon 1-R: 5′-CTTGGATCGAGGAAGTGACTCT-3′
*CFAP52* exon 2-F: 5′-TAGAAAATTCCTGGCCACTTCT-3′
*CFAP52* exon 2-R: 5′-CTGCAACCTCGGGTATAAAATC-3′
*CFAP52* exon 3-F: 5′-GTTTTTCACCAAGAAGGGTAAC-3′
*CFAP52* exon 3-R: 5′-CGTCTGAACCTGAAATAAAAGTTAAA-3′
*CFAP52* exon 4-F: 5′-CCACTAAGATGGTTGTTTCTTATTTT-3′
*CFAP52* exon 4-R: 5′-TTCAGCAGCAATCACAGAATTT-3′
*CFAP52* exon 5-F: 5′-GCCTCAAAAATGTTTAGATCTCTTC-3′
*CFAP52* exon 5-R: 5′-TGAGCTTCTTCAATCAACATCC-3′
*CFAP52* exon 6-F: 5′-TGCAATGGCACAATCTTGAC-3′
*CFAP52* exon 6-R: 5′-GTACATGCTGCCAGTGAAGG-3′
*CFAP52* exon 7-F: 5′-GGCCATCTTTAATTCATCTTTCA-3′
*CFAP52* exon 7-R: 5′-AGCCACCATCTATTGTGATTCC-3′
*CFAP52* exon 8-F: 5′-TGCCTTTCCAACTGTGTCACTT-3′
*CFAP52* exon 8-R: 5′-TTTCCATCAAGGGTTTATTAGAA-3′
*CFAP52* exon 9-F: 5′-TTGGTGTCAAATATGCATCTGG-3′
*CFAP52* exon 9-R: 5′-CCAAAGAGAGGCTAGATTAGAAGC-3′
*CFAP52* exon 10-F: 5′-AGCTGAGCTGGTCTCTTTCCT-3′
*CFAP52* exon 10-R: 5′-CCTGCTTGAATACTAGCAAAGGT-3′
*CFAP52* exon 11-F: 5′-CACAAATCAAGCATAGCAAAGG-3′
*CFAP52* exon 11-R: 5′-CCCACTAGCATCGATAAACCAT-3′
*CFAP52* exon 12-F: 5′-ACAAGCACCACTTTGCTGAG-3′
*CFAP52* exon 12-R: 5′-CCACTTAACAGGTGGGGAAG-3′
*CFAP52* exon 13-F: 5′-ATCCAGGTTTATCAGCACATCA-3′
*CFAP52* exon 13-R: 5′-CTGCAGTATTGGTTGGAATCTG-3′
*CFAP52* exon 14-F: 5′-AAAGACAGATACATGGGGACCA-3′
*CFAP52* exon 14-R: 5′-TGCAGAGTTGCAGAAAACAAAT-3′

The *CFAP52* reference sequence used here differs from the report by Ta-Shma et al., which used ENST00000352665.9 (Ensembl)/NM_145054 (NCBI) and encodes the 620 amino acid isoform (NP_659491).

**cDNA cloning and recombinant protein expression.** Full-length (551 amino acid, 66 kDa) human *CFAP45* was cloned from cDNA prepared from healthy nasal epithelial cells obtained by brush biopsy using the RNeasy Plus Kit (Qiagen). Nested PCR products were amplified using KOD DNA Polymerase (EMD Millipore) and subcloned into the pDONR201 Gateway entry vector (kindly provided by R. Roepman) using the following primers:

*CFAP45* 5′UTR: 5′-CGGCAGCAGAAGTCCGGAGTCAGGGCG-3′,
*CFAP45* 3′UTR: 5′-GTTAGCAGCAATTATGGATGCCCAGAGACTGGGC-3′,
*CFAP45* pDONR201-F:
5′-GGGGACAAGTTTGTACAAAAAAGCAGGCTTCATGCCACTAAGCACAGCTGGCATC-3′,
*CFAP45* pDONR201-R:
5′-GGGGACCACTTTGTACAAGAAAGCTGGGTTTAGTTCACAGAGGTAGCTGGCAG-3′

*CFAP45* was subsequently transferred via recombination into a Gateway destination vector (also kindly provided by R. Roepman) for recombinant protein expression containing an amino-terminal FLAG-epitope. DNA sequencing of this construct confirmed an exact match for *CFAP45* NCBI accession number NM_012337.2. We suspected that a hypothetical sequence encoding a 223 amino acid, 25 kDa protein (Aceview, NCBI: Mouse 2007, mRNA variant cSep07; GenBank: CD363487; MGI: 3182447) represents an isoform of CFAP45 (Fig. 1j, orange arrow). We cloned this isoform by nested PCR from mouse tracheal cDNA using the following primers:

*Cfap45* 5′UTR: 5′-GCCTGCAGCTGAAGCGGCCG-3′,
*Cfap45* 3′UTR: 5′-CCTATCCACGCGAGCCCCGC-3′,
*Cfap45* pDONR201-F:
5′-GGGGACAAGTTTGTACAAAAAAGCAGGCTTCATGCCACTAAGGCCAGGGGATGCCTC-3′,
*Cfap45* pDONR201-R:
5′-GGGGACCACTTTGTACAAGAAAGCTGGGTTTATCAGTCCCTCTCCTGGTCCTGGGTC-3′

This PCR product was subcloned into the myc-, FLAG-, and NTAP-epitope Gateway destination vector as described above.

We cloned *CFAP52* from healthy control human respiratory cDNA using a nested PCR approach using the following forward (F) and reverse (R) primers:

5′UTR *CFAP52*-F: 5′-GAGAGCAAAGTAATCAGAACCTCCCAAGG-3′
3′UTR *CFAP52*-R: 5′-GAAGACTAGTGTTGGAGTTATCTAAACTCAGTC-3′
pDONR201 *CFAP52*-F:
5′-GGGGACAAGTTTGTACAAAAAAGCAGGCTTCATGGATAACAAAATTTCGCCGGAGGCCC
pDONR201 *CFAP52*-R:
5′-GGGGACCACTTTGTACAAGAAAGCTGGGTTTAGGAGGTATATGGGTACTTCCATCGC.

This identified a transcript corresponding to ENST00000396219.7 (Ensembl)/NM_001080556 (NCBI) that encodes a 552 amino acid protein (NP_001074025). *CFAP52* was subsequently subcloned into the myc-, FLAG-, and NTAP-epitope Gateway destination vectors as described above. We confirmed this CFAP52 isoform by IB human ALI and human sperm lysates (Supplementary Fig. 5g, h).

High-resolution IFM analysis of human respiratory cells and mouse trachea samples were performed as described[33]. In Fig. 1, anti-CFAP45 immunofluorescence exposure time averaged <3 ms for controls and >15 ms for individuals

OP-28 II1 and OP-985 II1. In Fig. 3, anti-CFAP45 immunofluorescence exposure time averaged < 3 ms for DNAH5-, DNAAF1-, and CCDC40-mutant as well as control cilia and >15 ms for *DNAH11*-mutant cilia. In Supplementary Fig. 5, anti-CFAP52 immunofluorescence exposure time averaged <3 ms for controls and >15 ms. for individuals OI-81, OI-142, and OI-161. In Supplementary Fig. 8, anti-CFAP45 immunofluorescence exposure time averaged <3 ms for OI-161 and >15 ms for OI-140 and OI-142-mutant cilia; anti-IQCD immunofluorescence exposure time averaged >15 ms for OI-161, OI-140, OI-142, and OI-161. Anti-CFAP45 antibody (3618), anti-CFAP52 antibody (3247), anti-IQCD antibody (8782), anti-AK8 antibody (1445), and anti-DNAH17 antibody (4354) were used at a final concentration of 0.20 ng/μl for IFM of human respiratory cells; anti-CFAP45 and anti-CFAP52 were used at a final concentration of 0.33 ng/μl for IFM of mouse and pig respiratory cilia.

Briefly, porcine (*Sus scrofa*) respiratory cells isolated by brush biopsy were spotted on glass slides, dried overnight, and processed by IFM per published protocols. Freshly dissected tracheas from healthy adult pigs were washed thoroughly in 1× PBS, and brushed respiratory cells were collected in RPMI media to preserve viability. Respiratory cell suspensions were washed several times with 1× PBS followed by centrifugation (400 × *g* for 5 min. at 7 °C) to remove mucus and extracellular debris. Our standard IFM protocol for respiratory cells utilizing skim milk for blocking and antibody dilutions was adapted for IFM of sperm in a previous study[70]. We have recently optimized a protocol for sperm IFM that utilizes bovine serum albumin (BSA) in place of skim milk and additional washing steps to reduce false positive immunoreactivity.

Briefly, human and mouse sperm were diluted to ~$2 × 10^4$ sperm cells/50 μl, spotted on positively charged glass slides, dried overnight, and stored at −80 °C until processing for IFM. Before IFM, slides were defrosted, washed at least two times with 1× PBS and incubated in 10 mM citrate buffer at 99 °C for antigen retrieval. After cooling slides in 1× PBS, cells were fixed with 4% paraformaldehyde and then permeabilized with 0.1% Triton X-100 (diluted in 1× PBS) three times for 5 min. each. Cells were blocked with 1% BSA/0.1% Triton X-100 (diluted in 1× PBS) for 2 h at room temperature. Cells were then incubated overnight at 4 °C with primary antibody diluted in blocking solution. On the following day, cells were washed three times for 15 min each with 0.1% Triton X-100 diluted in 1× PBS, and then incubated overnight at 4 °C for 1 h at room temperature and protected from light with secondary antibody diluted in blocking solution. Cells were subsequently rinsed with 0.1% Triton X-100 diluted in 1× PBS and then with 1× PBS to remove unbound antibody and reduce nonspecific background signal. Cellular DNA was labeled using Hoechst 33342 diluted in 1× PBS, cells then washed with 1× PBS before mounting with DAKO antifade medium for preservation. Images were obtained using either a Zeiss LSM 880 laser scanning microscope with Zen-2 software or Zeiss AxioObserver Z1 Apotome with AxioVision 4.8 software (Carl Zeiss) and processed using Adobe CS4 software (Adobe Systems Incorporated); maximum intensity projections were generated using Zen-2 software. High-resolution Airyscan imaging to colocalize DNAH11 with CFAP45, CFAP52, and AK8 (Fig. 6) was performed using a Zeiss LSM 800 laser scanning microscope equipped with a 63× (n.a. = 1.4) objective lens. Colocalization coefficients were determined using Zen-2 software. For every IFM, a minimum of 12 images from at least two separate and blinded experiments were obtained.

**Antibodies and reagents**. The following rabbit polyclonal antibodies were purchased from Atlas Antibodies: anti-CFAP45, HPA043618; anti-DNAH11, HPA045880; anti-CFAP52, HPA023247; anti-IQCD, HPA048782; anti-DNAI1, HPA021649; anti-DNAH9, HPA052641; anti-DNAH5, HPA037470; anti-AK8, HPA021445; anti-DNAH17, HPA024354. Polyclonal anti-DNAH5 (amino acids 42–325) and monoclonal anti-DNALI1 antibodies were reported[66,70]. Mouse monoclonal anti-acetylated α tubulin (T7451) and Hoechst 33342 (B2261) were purchased from Sigma-Aldrich. Mouse monoclonal anti-DNAH11 (non-commercial, undiluted hybridoma supernatant) was described[33]. The following secondary antibodies were purchased from Dianova for IFM and used at a 1:200 dilution: anti-rabbit Rhodamine Red-X (711-295-152), and anti-mouse Cy3 (715-166-150). Anti-mouse Alexa Fluor 488 (A11029), anti-mouse Alexa Fluor 546 (A11030), anti-rabbit Alexa Fluor 488 (A11034) and anti-rabbit Alexa Fluor 546 (A11035) were purchased from Invitrogen and used at a 1:200 and 1:1000 dilution for human and mouse samples, respectively. Appropriate controls including secondary antibodies alone were performed. Normal rabbit (sc-2027) IgG (Santa Cruz Biotechnology Incorporated) was used as control for immunoprecipitations. HRP-linked anti-rabbit (NA934V) and anti-mouse (NA931V) antibodies (GE Healthcare) were used for IB.

HVMA analyses of ciliary and flagellar beating were performed as described[24,33,66].

Human sperm analyses of individual OP-28 II1 and healthy donors were performed in accordance with the standards set by the Declaration of Helsinki. Samples of human semen were obtained from healthy donors and individual OP-28 II1 with their prior written consent, under approval of the institutional ethical committees of the medical association Westfalen-Lippe and the medical faculty of the University of Münster (4INie). Per World Health Organization guidelines, asthenospermia is defined as reduced sperm motility (<32% of sperm are progressively motile). We determined that only 7% of sperm from individual OP-28 II1 were progressively motile, whereas 75% of these sperm displayed

non-progressive motility with circular/rotational movements or more rarely, 18% showed non-progressive motility without circular/rotational movements. Sperm were purified using the 'swim-up' method in human tubal fluid (HTF+) containing (in mM): 72.8 NaCl, 4.69 KCl, 0.2 $MgSO_4$, 0.37 $KH_2PO_4$, 2.04 $CaCl_2$, 0.33 Na-pyruvate, 21.4 lactic acid, 2.78 glucose, 25 $NaHCO_3$, and 21 HEPES, pH 7.35 (adjusted with NaOH) Briefly, liquefied ejaculate was overlaid with HTF+ and incubated for 1 h at 37 °C. The supernatant containing motile sperm was collected, washed and resuspended to $10 × 10^6$ sperm/ml in HTF++ medium supplemented with 3 mg/ml human serum albumin. Videos of human sperm motility (Supplementary Videos 1 and 2) are shown at original recording speed of 120 frames per second (fps). Manual tracking of sperm flagellar waveforms (Fig. 1e, f) was performed by first exporting single frames from high-speed microscopy videos (120 fps) of sperm cells from individual OP-28 II1 and healthy control individuals. From each tenth frame, the sperm flagella were traced manually by using Adobe Illustrator graphics software. The single flagellar tracks were then aligned to the longitudinal axis of the sperm cell head.

The swimming trajectories and motility of sperm from individual OP-28 II1 were compared to healthy control in low and high viscosity media (Supplementary Fig. 2). Briefly, dark field videos (one second at 80 fps) of spermatozoa ($1 × 10^6$ sperm/ml) were recorded in glass chambers (depth of 80 μm) using an Olympus microscope equipped with a heated stage set to 37 °C. Videos were analyzed in ImageJ (NIH, Bethesda, USA) using an open-access CASA plugin developed by Wilson-Leedy and Ingermann[71] that was adapted to human sperm. Sperm with circular trajectories were defined using established CASA parameters, having an VAP > x μm/s and a path straightness (STR) < 0.5. The following values were used for the plugin: VAP > 15 μm/s; STR < 2/π. Sperm analysis was not available for individual OP-985 II1, individual TB-19 II1, and individual OI-81.

TEM analyses of human, mouse and Schmidtea ciliary samples were performed as described[33,39]. TEM analysis was not performed for individuals OP-985 II1 and TB-19 II1.

**Mouse experiments**. All mouse experiments were approved by local government authorities (AZ84-02.05.20.12.163; 84-02.05.20.12.164, Landesamt für Natur, Umwelt und Verbraucherschutz Nordrhein-Westfalen, University Hospital Muenster; FBS-12-019, Osaka University; AH28-01, RIKEN Center for Developmental Biology) and complied with ethical regulations.

**Generation and analysis of CRISPR-mediated *Cfap45*−/− mice**. *Cfap45*−/− mice were generated using the CRISPR-Cas9 system. Small guide RNAs (sgRNAs) designed to delete *Cfap45* exon 3 were produced by in vitro transcription using the MEGA short script T7 kit (Ambion, AM1354). Capped synthetic mRNA for Cas9 was transcribed from the Cas9/pSP64T vector using the SP6 mMessage mMachine Kit (Ambion, AM1340). The Cas9 and sgRNAs were injected into C57BL/6 fertilized eggs as described[72]. The genotypes of *Cfap45*−/− pups were confirmed by polymerase chain reaction (PCR) and subsequent sequence analysis. Additional information including schematic overview and targeting sequences as well as PCR primer sequences are shown in Supplementary Fig. 7.

The frequency of viable *Cfap45*−/− offspring was ~25%, suggesting that homozygous mutants did not perish during development. We analyzed 28 *Cfap45*−/− homozygotes and the following phenotypes were scored: (a) LRA abnormalities (Fig. 4g) including dextrocardia (8.3%), reversed lung lobation (16.7%), aortic arch on right side (8.3%), azygos vein on right side (20.8%), stomach on right side (12.5%), abnormal liver lobation (25%), vena cava located to the left of aorta (29.2%); (b) hydrocephalus (82.4%); (c) small body size (57.7%). Four embryos from both *Cfap45* heterozygous and *Cfap45*−/− lines were analyzed for LRO ciliary beating (Fig. 4h) and LRO ciliary flow by PIV analysis (Fig. 4i). See also Supplementary Videos 3 and 4. Eight *Cfap45*−/− mice died before postnatal day 60 (P60); three males (P40, P42, and P59) and three females (P41, P42, and P47) showed hydrocephalus.

For analysis of mouse sperm motility (Fig. 4o) and mouse sperm velocity (Fig. 4p), the cauda epididymis of adult mice were recovered, immersed in warm HTF medium, and squeezed to let sperm swim out. Sperm were then capacitated by placing the suspension in an incubator (37 °C, 5% $CO_2$). Sperm were diluted to a suitable concentration (10× to 50×), and 5 μl of sperm suspension was placed in a sperm count chamber (SC 20-01-02-B, Leja, Netherlands). Motility of sperm was observed with an Axioplan 2 microscope (Zeiss) using a 20× lens for sperm velocity analysis and a 40× lens for sperm flagellar observation. The microscope stage was prewarmed at 37 °C with a glass plate heater (TP-S-E, Tokai Hit, Japan). Sperm motility was recorded with a high-speed camera (HAS-500; Directed, Tokyo, Japan) at 200 fps. Sperm velocity was analyzed with ImageJ (NIH), and traces of sperm flagella were drawn with sperm motion analyzing software Bohboh (BohBoh Soft, Tokyo, Japan). Videos of mouse sperm motility (Supplementary Videos 5 and 6) were recorded at 200 fps; movies play at 30 fps. Sperm from six males (three *Cfap45*+/− and three *Cfap45*−/− males) were analyzed in three independent experiments.

For rotational analysis of LRO ciliary beating, embryos were recovered and the region containing the LRO was then excised. The node cavity was filled with phenol red-free DMEM supplemented with 10% fetal bovine serum. The LRO was observed with the upright microscope (Axioplan 2, Zeiss), using a ×100 objective lens. The rotation of LRO cilia was recorded for 2 s (100 fps) with the use of a high-speed camera (HAS-500; Directed, Tokyo, Japan). Images were processed and analyzed with ImageJ (NIH). Nodal flow (Fig. 4i) was observed by multipoint

scanning confocal microscopy and particle image velocimetry (PIV) analysis as described[73]. Briefly, recovered embryos were cultured at 5% $CO_2$/37 °C in DMEM supplemented with 75% rat serum for 30 min. The region containing the LRO was then excised, and the LRO cavity was filled with DMEM supplemented with 10% fetal bovine serum and fluorescent microbeads (0.2 mm in diameter) with excitation and emission wavelengths of 505 and 515 nm, respectively (Invitrogen). The motion of the beads was monitored in planes of +5 and +10 μm relative to the cavity for 10 s (21 fps) with the use of a CSU-W1 confocal unit (Yokogawa) and an iXon-Ultra EMCCD camera (Andor Technology) connected to IX83 microscope (Olympus) equipped with ×60 objective lens. Time-series images for PIV analysis were captured at a resolution of 512 by 512 pixels and were processed with interrogation windows of 16 by 16 pixels with 50% overlap, corresponding to a spatial resolution of 4.3 by 4.3 mm. The time-averaged velocity distributions were calculated for 10 s intervals. Analysis was performed on four heterozyous (88 cilia, 5.2 rotations/s) and four null embryos (74 cilia, 2.4 rotations/s).

IFM and ISH analysis of mouse embryonic node samples were performed as described[29,33]. Motile monocilia of the mouse LRO are exclusive to pit cells, which express DNAH11; they are distinguished from non-motile LRO cilia of crown cells, which are located peripherally to pit cells and do not express DNAH11[33]. Briefly, control and $Cfap45^{-/-}$ mice were sacrificed and their organs dissected in 1× PBS, embedded in Shandon Cryomatrix (Thermo Fisher Scientific), frozen on dry ice and cut into 25-μm sections using a CM 3050 s cryostat (Leica). For ISH and IFM analysis of CFAP45, CD-1 wild-type mice were mated and females were checked for plugs the following morning. The morning of plug detection was defined as embryonic day (E) 0.5. Plug-positive mice were sacrificed at 8.25 dpc and embryos dissected in 1× PBS for further analysis. For whole-mount ISH, dissected embryos were fixed in 4% paraformaldehyde/1× PBS overnight at 4 °C, transferred to methanol the following day and stored at −80 °C until use. To generate ISH probes, a 573 base pair fragment of mouse $Cfap45$ (NM_027972.1) was cloned into the pCRII-TOPO vector (Thermo Fisher Scientific) by PCR with the following primers and used to generate a probe for hybridization:

$Cfap45$ ISH-F: 5′-ATCCGGAAAACTCTCAGTGCT-3′,
$Cfap45$ ISH-R: 5′-AAAGGAAACGGAGGGAGGAG-3′.

Purified plasmid was linearized with BamHI (sense) or Not1 (antisense) restriction enzymes (New England Biolabs); sense and antisense probes were generated by SP6 and T7-mediated transcription according to the manufacturer's directions (Roche).

### RNA interference (RNAi) and locomotion analysis in *Schmidtea mediterranea*.
A clonal line of asexual *S. mediterranea* (BCN-10, originally provided by E. Saló) was maintained at 20 °C in Montjuïch salts solution as described[74]. RNAi-based depletion of *Smed-cfap45* and *Smed-cfap52* was performed by injecting double-stranded RNAs (dsRNAs) into the intestinal cavity on 3 consecutive days per week for 2 weeks[74]. dsRNAs were synthesized through in vitro transcription of PCR products generated using the following primers:

*Smed-cfap45*:
5′-ATGAACTTCTGGCTGCTTCG-3′,
5′-GTTTTGTCTGATTTTCGAGCG-3′;
*Smed-cfap52*:
5′-GCACATGCATTATTTGGTGT-3′,
5′-TCCGTCATAAACTTCCCAAT-3′;

Locomotion assay of planarians was performed as described[39] in media containing 0.6% low-melting agarose. Each experiment was performed with 7–15 animals and the distances traveled were recorded as technical replicates on 3 different days. In Fig. 5b, the average travel distance measured in three independent experiments was normalized to the average distance measured for control (*gfp*) planarians (relative distance). Knockdown efficiency of *Smed-cfap45* and *Smed-cfap52* (Fig. 5a) relative to *Smed-gapdh* (relative expression) were determined to exceed 90% of control levels by quantitative PCR (qPCR) using the following primers:

*Smed-cfap45* qPCR-F: 5′-GCAATTAAAACTCAGAAAGATGAGG-3′,
*Smed-cfap45* qPCR-R: 5′-TCATTTCTTGTCGTCGTCCA-3′,
*Smed-cfap52* qPCR-F: 5′-TTCTCCGAATGAGAAATATCTGG-3′,
*Smed-cfap52* qPCR-R: 5′-AGATGACAAACAGCCGTCGT-3′.

The reference sequence for *Smed-cfap45* is KY523071 and the reference sequence for *Smed-cfap52* are pending submission.

### Cell culture of primary respiratory epithelial cells on air-liquid interface (ALI).
Human respiratory epithelial cells were obtained by nasal brush biopsy of healthy donors and cultured in collagen-coated tissue flasks. Upon reaching confluency, cell layers of primary respiratory epithelial cells were detached from the cell culture flask using collagenase type IV (200 U/mL; Worthington Biochemical Corporation) and cells were trypsinized for 5 min in 1× Trypsin-EDTA Solution (Sigma-Aldrich). ALI inserts (Costar™ Corning® 3470 Transwell™ Clear Polyester Membrane Inserts For 24-Well Plates, Corning Incorporated) were collagen-coated using rat-tail collagen in acetic acid (1:5). Respiratory epithelial cells were resuspended in B-ALI Growth Basal Medium (Lonza) and seeded to collagen-coated transwells (100,000 cells per insert). Cells were cultured at 37 °C in a humidified, 5% $CO_2$ incubator and fed with B-ALI Growth Basal Medium from the basolateral and apical compartment for up to 5 days. Air-lift of the ALI culture was performed by aspirating the medium of the apical compartment and replacing medium of the basal compartment by B-ALI differentiation medium. Medium was replaced three times per week.

### Immunoprecipitation and IB.
For immuno-affinity capture of native ciliary complexes, we recovered respiratory epithelial cells of porcine trachea using the brush biopsy technique. This represented a mixture of intact mucus-secreting and multi-ciliated cells as well as "detached" cilia, per observations by IFM. This mixture, referred to as isolated PRC, was washed several times with 1× PBS until supernatants were clear, and cell pellets were stored at −80 °C until use. PRC samples were lysed in the following buffer (50 mM Tris HCl, pH 8.0, 1% Igepal CA630, 10% glycerol, 0.5 mM EDTA, 150 mM NaCl) supplemented with Complete Ultra protease inhibitor and Phosphatase Iinhibitor Cocktail 2 (P5727) and Cocktail 3 (P0044) (Sigma-Aldrich). Lysates were snap-frozen in liquid nitrogen and homogenized for 2.5 min at 2000 rpm using a Mikro-Dismembrator unit (Sartorius). Lysates were centrifuged at $14,000 \times g$ for 20 min at 4 °C, and supernatants were filtered (0.22 μm pore size) and subsequently quantitated (typically between 1.5 and 3.0 mg/ml) using Pierce BCA Protein Assay Kit (Thermo Fisher Scientific). PRC lysates were split equally and immunoprecipitated with species-matched control antibody and either rabbit polyclonal anti-DNAH11 or rabbit polyclonal anti-CFAP45 overnight at 4 °C. Immunoprecipitates were collected using μMACs protein A micro beads and separator according to manufacturer's directions (Milteyni Biotec). Bound immunoprecipitates were washed five times with 200 μl of lysis buffer and one time with 100 μl of low-salt buffer (20 mM Tris HCl, pH 7.5) and subsequently eluted from columns in 200 μl of buffer containing 200 mM glycine, pH 2.5. Eluates were further precipitated using methanol and chloroform (protocol available upon request), and dried pellets were either directly resuspended in 60 μl of 1× sample buffer (15 μl of 4× NuPAGE LDS sample buffer; 39 μl of respective lysis buffer; 6 μl 1 M 1,4-dithiothreitol (DTT)) for IB or processed for LC/MS–MS (MS) analysis. Approximately 15 mg lysate was used per immunoprecipitation with a ratio of 1 μg antibody/500 μg lysate. Lysate is ~30 μg protein and immunoprecipitates represent 12.5 μl of a 50 μl pellet resuspension volume (Fig. 2).

For IB human and mouse sperm lysates as well as human ALI lysates (Fig. 1 and Supplementary Fig. 5), samples were prepared as described above using homogenization, centrifugation, and concentration by methanol–chloroform precipitation. Human and mouse sperm obtained from healthy donors or B6 wild-type mice were collected in RPMI media or 1× PBS, respectively, centrifuged at $400 \times g$ for 5 min at 4 °C and subsequently washed several times in 1× PBS with repeat centrifugation steps. Dried protein pellets were directly suspended in 1× sample buffer (described as above) and briefly heated at 70 °C. Sample viscosity was reduced by repeated pipetting and fine-gauge needle aspiration. Lysates were estimated at 2–4 μg/μl based on comparative silver stainings with lysates used for immunoprecipitations.

HEK293 cells were cotransfected with plasmids encoding either myc-DNALI1 and FLAG-CFAP45 or myc-CFAP45 and FLAG-DNALI1 using Gene Juice transfection reagent (IMD Millipore) according to the manufacturer's directions at a final concentration of 0.1 μg DNA/ml per construct. Following incubation of transfected cells at 37 °C for 24 h, HEK293 cells were detached from 100 mm cell culture plates by washing with 1× PBS, briefly centrifuged for 5 min at $400 \times g$, and lysed in 1.0 ml of buffer (50 mM Tris HCl, pH 8.0, 1% Igepal CA630, 10% glycerol, 0.5 mM EDTA, 150 mM NaCl) supplemented with Complete Ultra protease inhibitor and Phosphatase Inhibitor Cocktail 2 (P5727) and Cocktail 3 (P0044) (Sigma-Aldrich). Lysates were processed as described above and incubated with species-matched control antibody and either rabbit polyclonal anti-CFAP45 or polyclonal anti-DNALI1 (HEK293) overnight at 4 °C. Immunoprecipitates were collected using μMACs protein A micro beads and separator according to the manufacturer's directions (Milteyni Biotec). Bound immunoprecipitates were washed four times with 200 μl of lysis buffer and one time with 100 μl of low-salt buffer (20 mM Tris HCl, pH 7.5) and subsequently eluted from columns in 200 μl of buffer containing 200 mM glycine, pH 2.5. Eluates were further precipitated using methanol and chloroform (protocol available upon request), and dried pellets were directly resuspended in 60 μl of 1× sample buffer (15 μl of 4× NuPAGE LDS sample buffer; 39 μl of respective lysis buffer; 6 μl 1 M 1,4-DTT). Approximately 3 mg (HEK293) of lysate was used per immunoprecipitation and a ratio of 1 μg antibody/500 μg lysate was used for immunoprecipitations. HEK293 cells (CRL-1573) were purchased from ATCC (LGC Standards GmbH)

Samples were electrophoresed in NuPAGE Novex 3-8% Tris-Acetate (DNAH11 detection) or Novex 4–12% Bis–Tris gels (CFAP45 and DNALI1 detection) in prechilled 1× Tris Acetate or 1 × 2-(N-morpholino)ethanesulfonic acid running buffer, respectively, and transferred to PVDF filters (Invitrogen). Briefly, filters were incubated overnight at 4 °C with primary antibody, washed four times with TBS-T (7.5 min incubations), incubated with horseradish peroxidase (HRP)-conjugated secondary antibody (anti-rabbit light chain-specific) for 1 h at room temperature, and washed eight times with TBS-T (7.5 min. incubations) before processing for enhanced chemiluminescence with ECL Prime Western Blotting reagents (GE Healthcare). Molecular weights were estimated with Hi-Mark, SeeBlue Plus2 and Magic Mark protein ladders (Invitrogen). For 3–8% Tris-Acetate gels, 9 μl of Hi-Mark and 1 μl of Magic Mark protein standards were mixed; for 4–12% Bis–Tris gels, 4 μl of SeeBlue Plus2 and 1 μl of Magic Mark protein standards were mixed. For IB, the final concentrations of primary antibodies

(diluted in TBS-T buffer) were 25 ng/µl (rabbit anti-CFAP45 and rabbit anti-DNALI1) and 50 ng/µl (rabbit anti-DNAH11), respectively. Images were digitally acquired using a Peqlab FUSION-SL Advance Imager (VWR International) and modified for brightness and contrast using Adobe Photoshop CS4 (Adobe Systems Incorporated). TAP of recombinant protein was performed as described[75] but used 500 mM NaCl lysis buffer (Fig. 7c) as well as standard 150 mM NaCl lysis buffer (Supplementary Fig. 10e–g).

**Mass spectrometry**. Following immunoprecipitation and glycine elution, samples were subjected to methanol–chloroform precipitation and in-solution digest as described[75]. Analysis was performed on an Orbitrap Fusion Tribrid mass spectrometer (Thermo Scientific) coupled to an Ultimate3000 RSLCnano system (Thermo Scientific). Tryptic peptides were loaded at a flow rate of 30 µl/min in 0.1% trifluoroacetic acid onto a nano-trap column (300 µm i.d. × 5 mm pre-column, packed with Acclaim PepMap100 C18, 5 µm, 100 Å; Thermo Scientific). After 3 min, peptides were eluted and separated on the analytical column (75 µm i.d. × 25 cm, Acclaim PepMap RSLC C18, 2 µm, 100 Å; Thermo Scientific) by a linear gradient from 2% to 30% of buffer B (80% acetonitrile and 0.08% formic acid) in buffer A (2% acetonitrile and 0.1% formic acid) at a flow rate of 300 nl/min over 117 min. Remaining peptides were eluted by a 5 min. gradient from 30% to 95% buffer B and injected into the MS by nano-spray. A top speed method was used with a mass range from 335 to 1500 $m/z$. From a high resolution, MS pre-scan over a mass range of 500,335–201,500 was performed at a resolution of 70,000. Fragmentation by HCD was achieved at the ten most intense peptide ions (>200 counts) and were selected for HCD fragmentation using a normalized collision energy of 26. The resulting fragments were detected with a resolution of 120,000 in the ion trap at normal resolution. Every ion selected for fragmentation was excluded for 20 s by dynamic exclusion. The lock mass option was activated using a background mass of 445.12003 as described. Data analysis by Mascot (Matrix Science) and Scaffold (Proteome Software) was performed as described[76]. Specificity of associations was established as described[75] and included the identification of at least two unique peptides in test immuno-affinity purifications that did not appear in species-matched (rabbit) control IgG purifications. Peptides were further scrutinized by reciprocal BLAST to compare porcine to other species including human. True peptides specific for dynein ATPases such as DNAH11 showed 100% identity between porcine and human proteins, and false peptides that showed imperfect identity to paralogs such as DNAH9 and DNAH17 or ambiguous identity to multiple dynein proteins were excluded. Data are available via ProteomeXchange with identifier PXD019806.

**Yeast two-hybrid analysis**. The direct interaction between CFAP45 and 246 partial or full-length constructs from 175 "ciliaproteome" proteins (Supplementary Table 3) was tested using a GAL4-based (Hydrizap) yeast two-hybrid system (Stratagene) as described[77]. Full-length (551 amino acid) CFAP45 was used as a bait to test the interaction with ciliopathy and cilium-associated proteins fused to an activation domain (GAL4-AD). Constructs encoding GAL4-BD and GAL4-AD fusion proteins were co-transformed in yeast strain PJ69-4A. The direct interaction between baits and preys induced the activation of reporter genes, resulting in the growth of yeast colonies on selective media lacking both histidine (titrated with 5 mM 3-amino-1,2,4-triazole) and adenine as well as the induction of α-galactosidase and β-galactosidase visualized by colorimetric reactions. The standard controls for positive and negative yeast colony growth are pBD-USH2A_icd and pAD-NIN-L_isoB and pBD-USH2A_icd and pAD-GAL4, respectively.

**AMP binding assay**. Human sperm samples were collected from healthy, normospermic donors according to WHO guidelines. Sperm samples were washed several times with 1× PBS and centrifuged (1000 × g for 5 min, 4 °C) until supernatants were clear, and pellets were frozen at −80 °C until use. Sperm samples were lysed in high salt NP40 buffer (50 mM Tris-Cl, pH 8.0, 500 mM NaCl, 1% IGEPAL, 0.5 mM EDTA, 10% glycerol) supplemented with Complete Protease Inhibitor (Roche) and Phosphate Inhibitors Cocktails 2 and 3 (Sigma-Aldrich). Lysates were placed in a capsule with ball bearing, snap-frozen in liquid nitrogen, and homogenized using a Demembranator (2000 rpm for 2.5 min.). Homogenates were pooled and centrifuged at 14,000 × g for 20 min at 4 °C. Supernatants were further clarified using a syringe with 0.22 µM filter (Millipore). A lysis volume of 5.0 ml per 1.0 g of sample (w/w) typically yielded a concentration between 1.25 and 2.0 mg/ml. To remove free nucleotides prior to the AMP binding assay, sperm samples were processed with Zeba[TM] columns (Thermo Scientific) according to the manufacturer's directions.

To test if native CFAP45 protein could bind AMP, CFAP45 immunoprecipitates of sperm samples were incubated with AMP conjugated to agarose (Jena Bioscience, AK-107) according to the manufacturer's directions. Briefly, samples were precleared with control agarose resin (≥10 to 1 volume ratio of sample to packed agarose) by overhead rotation for 60 min at 4 °C and recovered by centrifugation (200 × g for 1 min at 4 °C). Samples were then equally distributed among four agarose resins (~5 ml lysate; 1.25 ml lysate to 125 µl packed resin) with AMP conjugated in different orientations (EDA-AMP agarose, 6AH-AMP agarose, αAH-AMP agarose, and 8AHA-AMP agarose) and incubated with overhead rotation overnight at 4 °C. Following sample centrifugation (200 × g for 1 min at 4 °C), supernatants with unbound proteins were removed, and AMP resins were transferred to illustra[TM]

MicroSpin Columns (GE Healthcare, 27-3565-01) by gently resuspending the resins with 2.0 ml of 1× TBS. Sample volume was allowed to flow passively through the columns and discarded; the columns retained AMP resin and bound protein. The columns were capped, filled with 1.0 ml of 1× TBS, mixed gently several times to resuspend the resin, and centrifuged (200 × g for 10 s at 4 °C) a total of three times to wash AMP resins. To elute AMP-bound proteins, the resins were incubated with 5′-AMP (Sigma-Aldrich, A1752) solution (100 mM) for 30 min on ice. Resins were mixed with the AMP solution by tapping the columns every 5 min. Eluates (~200 µl) were recovered by centrifugation (200 × g for 30 s at 4 °C), and dried protein pellets were obtained by methanol–chloroform precipitation for subsequent IB. With permission from Jena Bioscience GmbH (Germany), representations of agarose-conjugated AMP (Fig. 7a) have been adapted for this study.

**MT sliding assay**. Sperm from control and $Cfap45^{-/-}$ mice were analyzed for reconstitution of dynein ATPase activity (Fig. 7g, h) by the MT sliding assay as described[78,79]. Briefly, demembranated sperm were incubated with nucleotide (e.g., 1 mM ATP) and 33 mM 1,4-DTT. Disintegration of MTs was induced at 37 °C for 10 min, and the percentage of sperm with or without sliding were counted. Sperm obtained from the cauda epididymis were added to 50 µl of HTF medium with paraffin and incubated for 1 h at 37 °C, 5% CO₂. A 5 µl sample from this suspension was transferred to 100 µl demembranation solution (1 mM EDTA, 50 mM Hepes, pH 7.9, 0.2% Triton X-100), stirred gently for 30 s, and 10 µl of this suspension was transferred to 100 µl reactivation solution (1 mM EDTA, 1 mM ATP, 5 mM MgSO₄, 33 mM DTT, 50 mM Hepes, pH 7.9) and incubated for 10 min. A 5 µl sample was placed on a glass slide with a coverslip, and sperm with or without sliding were counted. HTF medium:101.6 mM NaCl, 4.69 mM KCl, 0.37 mM KH₂PO₄, 0.2 mM MgSO₄·7H₂O, 21.4 mM Sodium lactate, 0.33 mM sodium pyruvate, 2.78 mM D-(+)-glucose, 25 mM NaHCO₃, 2.0 mM CaCl₂·2H₂O, 0.2 mM penicillin G sodium salt, 0.3 mM streptomycine sulfate, 0.0002% phenol red, 4.00 g/liter BSA. The Fisher's exact test with multiple comparison correction (Holm's method) was used for statistical analysis. Sample size (number of sperm) and independent experiments (number of males) are provided within Fig. 7g, h.

**Dynein ATPase extraction and sucrose gradient fractionation of porcine tracheal cilia**. Dynein ATPases were extracted from porcine respiratory cilia as described[46]. Briefly, fresh porcine tracheal sections (6-8 cm) were incubated in a solution of 20 mM Tris-HCl (pH = 7.9), 50 mM NaCl, 10 mM CaCl₂, 1 mM EDTA, 7 mM 2-mercaptoethanol and 0.1% Triton X-100 to isolate cilia. This suspension was centrifuged for 2 min at 1500 × g, and the supernatant was further centrifuged for 10 min at 12,000 × g. The pellet containing cilia was resuspended in a small volume in buffer with 20 mM Tris-HCl (pH = 8), 50 mM KCl, 4 mM MgSO4, 1 mM DTT, 0.5 mM EDTA and 0.1 mg/ml soybean trypsin inhibitor and diluted further in dynein extraction buffer (600 mM KCl containing 0.1 mM ATP) for 30 min at 4 °C. The sample was centrifuged at 31,000 × g for 15 min and the supernatant enriched for dynein ATPases (~4 ml) was dialyzed for 2 h in buffer containing 50 mM NaCl, 1 mM DTT and 10 Mm Tris-HCl (pH = 7.5). Approximately 2.5 ml of supernatant was added to 9 ml of sucrose solution, and the suspension was subsequently fractionated across a 5–30% sucrose gradient at 153,000 × g for 16 h. Nineteen fractions of 600 µl were collected and analyzed by IB (Supplementary Fig. 9).

**Bioinformatics analysis**. Lists of gene symbols corresponding to peptides identified by MS in CFAP45 or CFAP52 but not control immunoprecipitates were analyzed for functional annotation with DAVID Bioinformatics Resources 6.7 using high stringency. Nucleotide-binding predictions for CFAP45, CFAP52, and all human dynein ATPases were analyzed using NSitePred[50].

**Theoretical model of the CFAP45·AMP·AK8 complex**. Different initial models of CFAP45 and AK8 were generated using the NCBI reference sequences of NP_036469.2 and NP_689785.1 by MUSTER[80] and Swiss-Model[81], respectively. The crystal structure of a thermally stable AK mutant (PDB ID: 4TYQ) was used as a template for homology modeling of AK8. Visual inspection of these models identified charge complementarity between Model 1 of the KD2 domain of AK8 and Model 2 of the Trichoplein domain of CFAP45. Using MacPyMOL (PyMOL v1.7.0.3 Enhanced for Mac OS X), we manually placed the Trichoplein domain in the vicinity of the KD2 domain to form a crude CFAP45·AK8 dimer, and then manually inserted AMP into a cavity at the interface of the dimer in such a way that AMP was primarily interacting with CFAP45. The resulting AMP·CFAP45·AK8 complex was energy-minimized and then substantially refined using two-step MD simulations. The RESP charges of AMP were obtained from the ab initio calculation using Gaussian 98 (revision A.7; Gaussian, Inc) with HF/6-31G*//HF/6-31G* according to a published protocol[82]. Topology and coordinate files of the AMP·CFAP45·AK8 complex were generated by the tLeAP module of AmberTools 13 (University of California, San Francisco). The energy minimization and MD simulations were performed using SANDER and PMEMD of AMBER 16 (University of California, San Francisco) with the FF12MC forcefield[52]. The energy minimization used dielectric constant of 1.0, and 100 cycles of steepest-descent minimization followed by 900 cycles of conjugate-gradient minimization. The energy-minimized AMP·CFAP45·AK8 complex was neutralized with a chloride and then solvated with TIP3P water molecules[83] and energy-minimized as above.

The resulting system was then heated from 0 to 340 K at a rate of 10 K/ps under constant temperature and volume, and finally simulated in 100 unique and independent all-atom, isothermal–isobaric, 63.2-ns MD simulations using a periodic boundary condition at 340 K and 1 atm with isotropic molecule-based scaling. All simulations used (1) a dielectric constant of 1.0, (2) the Berendsen coupling algorithm[84], (3) the Particle Mesh Ewald method to calculate electrostatic interactions of two atoms at a separation of >8 Å[85], (4) a time step of 1.0 fs of the standard mass time[52], (5) SHAKE-bond-length constraints applied to all the bonds involving the H atom, (6) a protocol to save the image closest to the middle of the "primary box" to the restart and trajectory files, (7) the revised halide ions parameters[86], (8) a nonbonded cutoff of 8.0 Å, and (9) the atomic masses of the entire simulation system (both solute and solvent) that were reduced uniformly by ten-fold[52], and (10) default values of all other inputs of the PMEMD module. The AMP·CFAP45·AK8 complex conformations were saved at 100-ps intervals in all simulations performed on 100 dedicated 12-core Apple Mac Pros with Intel Westmere (2.93 GHz). Cluster analysis of all saved AMP·CFAP45·AK8 conformations using the CPPTRAJ module of AmberTools 16 with the average-linkage algorithm[87] (epsilon = 2.0 Å; RMS on all CA atoms). The most populated conformation was used as the initial complex conformation for the second-round refinement using 100 unique and independent all-atom, isothermal–isobaric, 126.4-ns MD simulations, under the same simulation conditions as those of the first-round refinement. The most populated complex conformation (Fig. 7i) using the same cluster analysis protocol was used as the AMP·CFAP45·AK8 complex.

**Statistics and reproducibility**. Statistical analysis of data in Figs. 4h, o, p, 5a, b, 7g, h, and Supplementary Fig. 2c, d was performed with GraphPad Prism v.5.01. Sample size and number as well as statistical test used and $p$ values obtained for Figs. 4h, o, p, 5a, b, and 7g, h are provided within the main text and respective figure legends; statistical significance was not assessed in Supplementary Fig. 2c, d.

**Reporting summary**. Further information on research design is available in the Nature Research Reporting Summary linked to this article.

## Data availability

The data that support the findings of this study are available from the corresponding author upon reasonable request. WES data is available in a restricted manner due to privacy considerations and is contingent on the approval of the Institutional Ethics Review Board from the University Hospital Münster (Germany) and ethics committees from collaborating institutions. Proteomic data are available at a PRIDE Partner Repository via ProteomeXchange with identifier PXD019806. The following publicly available databases were used: gnomAD, https://gnomad.broadinstitute.org/; 1000 Genomes, https://www.internationalgenome.org/; MGI, http://www.informatics.jax.org/; GenBank, https://www.ncbi.nlm.nih.gov/genbank/; Uniprot, http://www.uniprot.org/; Ensembl, http://www.ensembl.org/index.html; Pfam, http://pfam.xfam.org/; Varbank, https://varbank.ccg.uni-koeln.de/; PROVEAN, http://provean.jcvi.org/index.php; Polyphen-2, http://genetics.bwh.harvard.edu/pph2/; DAVID Bioinformatics Resources 6.7, https://david-d.ncifcrf.gov/; Ciliome Database, http://www.sfu.ca/~leroux/ciliome_home.htm.; The Human Protein Atlas, http://www.proteinatlas.org/; NsitePred, http://biomine.cs.vcu.edu/servers/NsitePred/. Source data are provided with this paper.

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

## Acknowledgements

We thank the study individuals for participation, the German patient group 'Kartagener Syndrom und Primaere Ciliaere Dyskinesie e.V.', Holstiege Farm (Roxel, Münster), K. Wohlgemuth, A. Robbers, C. Westermann, B. Lechtape, M. Herting, D. Ernst, L. Schwiddessen, F. J. Seesing, S. Helms and C. Hartmann for technical support, A. Sabo, S. Dugan-Rocha, D. Muzny, and R. Gibbs for assisting targeted (ciliaproteome) NGS, R. Durbin and A.K. Kokocinski for WES, and Kyosuke Shinohara (Tokyo University of Agriculture and Technology) and Koji Okamoto (The University of Tokyo) for software of PIV analysis. We thank D.R. Mitchell for critically reviewing the manuscript and C.B. Lindemann for discussing insights on ADP. This research was supported by the generous support of the Victoria Popkin Fund through Hadassah, the Women's Zionist Organi-zation of America, Inc. to A.T., the Deutsche Forschungsgemeinschaft (DFG)  OM6/7, OM6/8, OM6/9, OM6/10, OM6/11, Interdisziplinaeres Zentrum für Klinische For-schung, Muenster IZKF (Om2/009/12 and Om/015/16), and BESTCILIA 305,404 grant to H.Omran, DFG CRU326 to T.S. and H. Omran, the European Commission FP7/ 2007–2013 grant 262,055 to Y.M. and H. Omran as a Transnational Access project of the European Sequencing and Genotyping Infrastructure, the European Union FP7 program SYSCILIA grant 241,955 to R.R., M.U., N.K. and H. Omran, the Netherlands Organi-sation for Health Research and Development (ZonMW, #91216051) to R.R., the CREST Japan Science and Technology Corporation 13417760 and Japan Society for the Pro-motion of Science 24113004 grants to H.H., the Max Planck Society EXC 1003-CiM (Cluster of Excellence Cells in Motion) and German Research Foundation BA4044/3-1 grants to K.B., and U.S. National Institutes of Health grant DK072301 to N.K. and E.E.D. N.K. is a distinguished George W. Brumley Professor.

## Author contributions

Study design and manuscript preparation: G.W.D. and H. Omran; clinical analysis: J.W., J.R., C.W., T.K., H.M., M.A., L.B., J.G.O., A.T., Z.P., O.E., I. Amirav and H. Omran; targeted NGS, WES and variant analysis: A.T., Z.P., O.E., Y.M., K.P., B.D., G.W.D., N.K. and E.E.D; Sanger sequencing: A.T., F.S., J.A.K., G.W.D. and H. Olbrich; semen analysis: L.B., S.Y., T.S. and I. Aprea; immunofluorescence microscopy: T.M., I. Aprea, F.S., J.A.K., I.M.H., R.H., N.T.L., P.P., J.G. and G.W.D.; in situ hybridization: T.M.; generation and analysis of *Cfap45*−/− mice: Y.I., K.T., H.S., W.K.T, T.I., K. Minegishi, K. Mizuno and H.H.; conventional transmission electron microscopy: P.P., T.M., I. Aprea and G.W.D.; Schmidtea analysis: F.R. and K. Bartscherer; immunoprecipitations and immunoblotting: G.W.D., F.S., I. Aprea, D.C.B. and S.C.; cDNA cloning: G.W.D., T.M., D.C.B. and F.S; yeast two-hybrid analysis: S.J.L. and R.R.; mass spectrometry analysis: K. Boldt, N.H., K.J. and M.U. Y.P.P. simulated the CFAP45·AMP·AK8 complex. P.P.D. contributed intellectually. All authors analyzed the data.

## Funding

## Competing interests

The authors declare no competing interests.

## Additional information

Gerard W. Dougherty[1], Katsutoshi Mizuno[2], Tabea Nöthe-Menchen[1], Yayoi Ikawa[2,3], Karsten Boldt[4], Asaf Ta-Shma[5], Isabella Aprea[1], Katsura Minegishi[2,3], Yuan-Ping Pang[6], Petra Pennekamp[1], Niki T. Loges[1], Johanna Raidt[1], Rim Hjeij[1], Julia Wallmeier[1], Huda Mussaffi[7,8], Zeev Perles[5], Orly Elpeleg[9], Franziska Rabert[10], Hidetaka Shiratori[3], Stef J. Letteboer[11], Nicola Horn[4], Samuel Young[12], Timo Strünker[12], Friederike Stumme[1], Claudius Werner[1], Heike Olbrich[1], Katsuyoshi Takaoka[2,3], Takahiro Ide[2], Wang Kyaw Twan[2], Luisa Biebach[1], Jörg Große-Onnebrink[1], Judith A. Klinkenbusch[1], Kavita Praveen[13], Diana C. Bracht[1], Inga M. Höben[1], Katrin Junger[4], Jana Gützlaff[1], Sandra Cindrić[1], Micha Aviram[14], Thomas Kaiser[1], Yasin Memari[15,16], Petras P. Dzeja[17], Bernd Dworniczak[18], Marius Ueffing[4], Ronald Roepman[11], Kerstin Bartscherer[10,19,20], Nicholas Katsanis[13,21,22], Erica E. Davis[13,21,22], Israel Amirav[23,24], Hiroshi Hamada[2,3] & Heymut Omran[1✉]

[1]Department of General Pediatrics, University Hospital Münster, Münster, Germany. [2]RIKEN Center for Biosystems Dynamics Research, Kobe, Japan. [3]Developmental Genetics Group, Graduate School of Frontier Biosciences, Osaka University, Osaka, Japan. [4]Institute for Ophthalmic Research, Molecular Biology of Retinal Degenerations and Medical Bioanalytics, University of Tübingen, Tübingen, Germany. [5]Department of Pediatric Cardiology, Hadassah-Hebrew University Medical Center, Jerusalem, Israel. [6]Department of Molecular Pharmacology and Experimental Therapeutics, Mayo Clinic, Rochester, MN, USA. [7]Schneider Children's Medical Center, Prtach-Tiqva, Israel. [8]Sackler Faculty of Medicine, Tel Aviv University, Ramat, Aviv, Israel. [9]Department of Genetics, Hadassah-Hebrew University Medical Center, Jerusalem, Israel. [10]Max Planck Research Group on Stem Cells & Regeneration, Max Planck Institute for Molecular Biomedicine, Münster, Germany. [11]Department of Human Genetics and Radboud Institute for Molecular Life Sciences, Radboud University Medical Center, Nijmegen, the Netherlands. [12]Centre of Reproductive Medicine and Andrology, University Hospital Münster, University of Münster, Münster, Germany. [13]Center for Human Disease Modeling, Duke University Medical Center, Durham, NC, USA. [14]Soroka Medical Center, Beer-Sheva, Israel. [15]Department of Medical Genetics, Addenbrooke's Hospital, Cambridge, UK. [16]MRC Cancer Unit, Hutchison / MRC Research Centre, Cambridge, UK. [17]Department of Cardiovascular Medicine, Mayo Clinic, Rochester, MN, USA. [18]Department of Human Genetics, University of Münster, Münster, Germany. [19]Medical Faculty, University of Münster, Münster, Germany. [20]Hubrecht Institute, Utrecht, the Netherlands. [21]Advanced Center for Translational and Genetic Medicine (ACT-GeM), Stanley Manne Children's Research Institute, Ann & Robert H. Lurie Children's Hospital of Chicago, Chicago, IL, USA. [22]Department of Pediatrics, Feinberg School of Medicine, Northwestern University, Chicago, IL, USA. [23]Pulmonology Unit, Dana-Dwek Children's Hospital, Tel-Aviv University, Tel-Aviv, Israel. [24]Department of Pediatrics, University of Alberta, Edmonton, Canada. ✉email: Heymut.Omran@ukmuenster.de

