## [Peer Review File · Nature Communications]

Reviewers' comments:

Reviewer #1 (Remarks to the Author):

The authors seek to understand the biologic basis of mutations in ciliary protein CFAP45 resulting in a motile ciliopathy. Proteomic profiling link CFAP45 to adenylate kinase AK8, axonemal dynein components of the inner and outer dynein arm complex, CFAP52 and ENKUR, also proteins associated with human motile ciliopathies. Model organisms with motile cilia phenotypes associate support that the mutations in CFAP45 and CFAP52 are pathogenic. Pull down experiments support that CFAP45 and 52 interact and also with AK8, while in vitro assays identify CFAP45-AMP binding. There remains some doubt as to the final proposed mechanism and model for regulation of ciliary motility relative to location of the CFAP45/CFAP52/AK8 complex. The recent paper by Ma et al., *Cell* Oct 31, 2019 (PMID: 31668805), clearly show that CFAP45 and CFAP52 are microtubule inner proteins (MIPs) in close proximity in the B-tubule, making it difficult to conceive how a protein complex in this region would bind and regulate DNAH11 in the outer dynein arm on the A tubule. The new findings are consistent with similar results noted in studies (Owa, et al., 2019) cited by the prior reviewers, but Ma's data are very strong evidence as to the location and it is doubtful that the mammalian structure is going to be radically different. This reviewer considers that one possibility is through B to A MT communication since Ma's data shows that ENKUR (FAP106), which Dougherty et al. IP with CFAP45, may link interactions in the B tubule with the A tubule. At this point, it would be speculative that such communication could control ciliary beating.

Another possibility for loss of regulation of cilia function in the CFAP45/CFAP52 mutants could be that the function of the NDR-C (which spans the A and B tubules nearby CFAP45 and 52) is disrupted or missing components since IQCD was in CFAP52 proteomics and located in the proximal part of the cilia.

Also, it is not shown if there are differences for the role of ADP for the inner and outer dynein arms. The data supporting the association of the DNAH11 to the mutant is not as specific or strong as that interactions for CFAP45-52. It is provocative that both outer and inner dynein arm proteins are in pull downs with CFAP52. Reference 11 and 12 demonstrate a role for ADP binding to inner dynein arm dynein a. Are their biochemical data related to IDA dyneins? Is there NsitePred analysis with proteins at the inner and outer dynein arms?

Reviewer #2 (Remarks to the Author):

The most significant finding of this manuscript is the identification of CFAP45 as a motile ciliopathy gene. The authors further suggest that CFAP45, CFAP52 and AK8 forms a functional module for ADP homeostasis and indirectly regulates dynein ATPases. Although the importance of ATP/ADP in cilia motility is known, this study provides a specific function for CFAP45. The model put forth by the authors is consistent with the localization of all three proteins to the axoneme and co-IP between them. More direct evidence is provided by the ability of CFAP45 to bind AMP and that microtubule sliding of cfap mutant sperms can be rescued by exogenously added ATP and ADP (Figure 7 and associated supplementary materials). However, this also is an area that needs to be strengthened if the main conclusion is that CFAP45 regulates ADP homeostasis through AK8. Moreover this set of results can be better presented. The statement "CFAP45 immunoprecipitates recovered after high salt lysis of human sperm samples showed affinity to AMP..." is puzzling. Does it mean that CFAP45 was pulled down by AMP beads? Or CFAP is IPed, eluded then pulled down by AMP beads? The results with recombinant CFAP45 and AK8 in fig 7c-e potentially suggest that CFAP45 binds to AMP directly. However, the possibility of non-specific binding to beads needs to be ruled out. More importantly, the activity and properties of AK8 has been described in *Biochem. J.*

(2011) 433, 527–534. The authors seem to be at a position to perform biochemical experiments with recombinant proteins to directly ask whether CFAP45 affect the activity of AK8. Results potentially can provide direct and convincing evidence for their model. It will also be interesting to test whether CFAP45 show binding specificity toward AMP and ADP, because an alternative model is that AMP or ADP binding induces a conformational change in CFAP45 that is unrelated to enzymatic conversion by AK.

I found fig. 7g and h difficult to follow. The authors state “Administration of 1mM ATP alone or 1mM ADP alone did not significantly reactivate MT sliding of *Cfap45*^{-/-} sperm compared to heterozygous control (Fig. 7g); additionally, 1mM AMP alone showed negligible (< 1%) reactivation of MT sliding in control or *Cfap45*^{-/-} sperm, similar to untreated control (Fig. 7g).” However, it looks to me that 1mM ATP or ADP alone did significantly reactivate MT sliding of *Cfap45*^{-/-} sperm--from almost none to more than 40%--although not to the same degree as heterozygous control. Similarly, 1mM AMP did not reactivate microtubule sliding in either mutant or control samples (fig. 7g). The key result seems to be that 1mM ATP with 4mM AMP or ADP rescues better than 1mM AMP or ADP in fig 7h. However, the difference is quite modest. In addition, the impact of 4mM ADP or AMP alone needs to be tested to reach a meaningful conclusion. Moreover, if the function of CFAP45 is to provide a platform for forward AK reaction toward ADP production, one would expect a differential response of *cfap45* mutant samples to ATP in combination with ADP vs AMP (fig 7h, compare the first two groups of columns). Separate but related, since *Cfap45*^{-/-} sperm can still beat but show abnormal curvature and amplitude, the impact of adenine nucleotides on mutant sperm beating pattern will be interesting to test.

The final and important piece of evidence is provided by structural modeling. I do not consider myself qualified to evaluate this approach. However, it will be helpful to provide some independent assessment. For example, if the motile cilium proteome is run through NSitePred, how many hits would the program pick up.

Regarding the title, “harmonize” at first glance hints at coordination between cilia, which does not seem to be the topic of this manuscript.

Supplementary table 3 is missing.

I also think the list of tested proteins in the 2 hybrid assay should be provided in supplementary materials, instead of “available upon request”. Negative results will be useful to the community and provide a context for the positive results presented in this manuscript.

We thank the editors for considering our manuscript and the reviewers for critically reviewing our manuscript. We appreciate the constructive feedback and believe that our responses with additional supporting data reasonably address these concerns and warrant publication in Nature Communications. Please see our responses to specific concerns below:

Reviewer #1

The authors seek to understand the biologic basis of mutations in ciliary protein CFAP45 resulting in a motile ciliopathy. Proteomic profiling link CFAP45 to adenylate kinase AK8, axonemal dynein components of the inner and outer dynein arm complex, CFAP52 and ENKUR, also proteins associated with human motile ciliopathies. Model organisms with motile cilia phenotypes associate support that the mutations in CFAP45 and CFAP52 are pathogenic. Pull down experiments support that CFAP45 and 52 interact and also with AK8, while in vitro assays identify CFAP45-AMP binding.

a) We appreciate this concise summary.

There remains some doubt as to the final proposed mechanism and model for regulation of ciliary motility relative to location of the CFAP45/CFAP52/AK8 complex. The recent paper by Ma et al., Cell Oct 31, 2019 (PMID: 31668805), clearly show that CFAP45 and CFAP52 are microtubule inner proteins (MIPs) in close proximity in the B-tubule, making it difficult to conceive how a protein complex in this region would bind and regulate DNAH11 in the outer dynein arm on the A tubule. The new findings are consistent with similar results noted in studies (Owa, et al., 2019) cited by the prior reviewers, but Ma's data are very strong evidence as to the location and it is doubtful that the mammalian structure is going to be radically different.

b) We appreciate the recent studies by Ma et al, 2019 (PMID: 31668805) and Owa et al, 2019 (PMID: 30850601) that describe CFAP45 (as well as associated proteins) as luminal proteins of the B-tubule. However, there is evidence suggesting CFAP45 has a functional role at the A-tubule where dynein ATPases are located. In the study by Owa, it is clear that single FAP45 (and also FAP52) Chlamy mutants show a reduction in *dynein-b* intensity at the A-tubule:

Fig. 2 Dynein b density is reduced in the *fap45* and *fap52* DMT. Isosurface renderings of averaged DMT repeats: wild type (a), *fap45* (b), and *fap52* (c). The density of dynein b is decreased in *fap45* and *fap52* (arrowheads). Other species of IDAs, ODAs, RS, and N-DRC were properly arranged on the DMT in *fap45* and *fap52*. Scale bar = 25 nm

However, those authors do not further address this phenotype and focus on characterizing the microtubule doublet defect using combined FAP45/FAP52 Chlamy mutants. Our data supports detection of CFAP45 (and also CFAP52) on the surface of the axoneme; this is partially explained by inherent constraints of immunofluorescence microscopy under our protocol conditions. The outer diameter of the microtubule doublet is 25 nm whereas the inner diameter is approximately 16 nm. A full length immunoglobulin species (heavy and light chain) is approximately 10 nm x 10 nm in dimension. We use intact (heavy and light chain) primary and secondary antibodies for detection, raising the effective working space for this antibody-antibody complex to at least 20 nm in diameter. The antibodies are only accessible to the outer surface of the axoneme unless completely disruptive conditions such as sarkosyl are used. If CFAP45 and CFAP52 are strictly luminal, we would not expect to robustly detect these proteins in the absence of sarkosyl or a similarly disruptive reagent. This is consistent with our data showing colocalization of CFAP45, CFAP52 and AK8 with DNAH11 in respiratory axonemes (without sarkosyl) by Airyscan high-resolution imaging. We provide additional evidence that CFAP45, CFAP52 and AK8 co-elute with DNAH11 following cilia isolation, high salt / ATP buffer extraction, and sucrose gradient fractionation. In turn, these data are consistent with observations that CFAP45 identifies DNAH11 by immunoprecipitation and LC/MS-MS or Western blotting as well as the loss of axonemal CFAP45 in respiratory axonemes of several unrelated, genetically distinct *DNAH11*-mutant respiratory samples. Future studies such as cryo-EM of CFAP45-deficient human and mouse cilia and flagella should help provide context for these results. However, we wish to stress that ultrastructural localization using these techniques was not the principal goal of our study but rather to characterize the molecular and genetic basis of novel motile ciliopathy cases. We have now included new data highlighting three additional *DNAH11*-mutant respiratory samples where CFAP45 shows accumulation at the ciliary base but not along the axoneme (please see new **Supplementary Figure 4**).

In contrast, CFAP45 shows pan axonemal immunoreactivity in three loss-of-function *DNAH9* respiratory samples comparable to healthy control samples. This represents an elegant control to ascertain potential association with outer arm dynein ATPases, as *DNAH11* and *DNAH9* are paralogs that have adapted functions in the proximal and distal compartments, respectively, within the same respiratory cilium.

While the conservation of biological function and structure between axonemes of blue-green alga and human respiratory cells is impressive, there are clearly structural and molecular differences that reflect evolutionary adaptations / modifications of these organelles, including the outer dynein arms. In *Chlamydomonas*, the outer dynein arm is a three-headed structure (α , β , and γ heavy chain ATPases) that has been reduced to a two-headed dynein arm structure (β and γ heavy chain orthologs, *DNAH11/9* and *DNAH5*, respectively) in mammals; the *Chlamy* α heavy chain has no clear mammalian ortholog. Furthermore, the *Chlamy* β heavy chain dynein gene has duplicated not once but twice to reflect tissue- and axoneme-specific functions (*DNAH11* and *DNAH9* in the proximal and distal ciliary regions of respiratory cilia, *DNAH17* in sperm flagella). We and others have shown that the β -ODA-HC paralog *DNAH11* behaves quite differently from the β -ODA-HC paralog *DNAH9*. Mutations in *DNAH9* disrupt distal ODA assembly (see below, Figure 4, Loges et al., 2018, PMID: 30471718), thus behaving more similarly to the ancient β -heavy chain dynein in *Chlamydomonas*. In contrast, mutations in *DNAH11* do not disrupt assembly of proximal ODAs (Dougherty et al., 2016).

This reviewer considers that one possibility is through B to A MT communication since Ma's data shows that ENKUR (FAP106), which Dougherty et al. IP with CFAP45, may link interactions in the B tubule with the A tubule. At this point, it would be speculative that such communication could control ciliary beating.

Another possibility for loss of regulation of cilia function in the CFAP45/CFAP52 mutants could be that the function of the NDR-C (which spans the A and B tubules nearby CFAP45 and 52) is disrupted or missing components since IQCD was in CFAP52 proteomics and located in the proximal part of the cilia.

- c) We originally focused on the common and reciprocal associations of AK8 to CFAP45 and CFAP52 rather than the unique associations of either CFAP45 to ENKUR or CFAP52 to IQCD as a potentially unifying mechanism. We have now tested ENKUR by IFM in CFAP45-mutant respiratory and sperm samples and determined that they are normal, similar to our observations with CFAP52 and AK8 (data not shown). We have also tested ENKUR-deficient respiratory cilia (patient described in Sigg et al., *Dev. Cell*, PMID: 29257953) and determined that both CFAP45 and CFAP52 are normal, comparable to wild type localization (data not shown).

We acknowledge your suggestion that the N-DRC may be disrupted in CFAP45 / CFAP52 deficiency and have explored this further. As we reference in the original submission, Bower et al. identified DRC10 (the *Chlamydomonas* ortholog to IQCD) in GAS8-immunoprecipitates from *Chlamydomonas* axonemes. We have examined IQCD in cilia from PCD individuals that

we previously reported with *CCDC40* (Becker-Heck et al., Nature Genetics, 2011) and *GAS8* (Olbrich et al., Am J Hum Genet, 2015) mutations that disrupt the N-DRC. Similar to proximal ciliary localization in healthy controls, IQCD is still detectable in the proximal region in cilia from individual OP-1186 with homozygous loss-of-function *CCDC40* mutations (p.Arg814* hom.) and individual OP-1940 with homozygous loss-of-function *GAS8* mutations (p.Arg334* hom.) (data not shown). As we show in the original submission, both CFAP45 and CFAP52 are detectable in these N-DRC-mutant cilia (Figure 3 g,h; Supplementary Fig. 6 f-h). Thus, the axonemal localization of these three proteins seems not to be affected by disruption of the N-DRC. More importantly, the correlation of situs abnormalities in CFAP45- and CFAP52-deficient individuals strongly suggests that the N-DRC is not appreciably affected. To date, no human mutations in genes encoding individual components of the N-DRC complex (*GAS8*, *CCDC164*, *CCDC65*) have been reported to cause *situs inversus totalis*, raising doubts that N-DRC components (including IQCD) have an essential role in human embryonic nodal cilia (9+0). Conversely, there is a very strong correlation between mutations affecting the dynein arms (including *DNAH11*) and PCD cases with *situs inversus totalis*. Our mechanistic interpretation incorporates these tissue-specific differences among respiratory (9+2), embryonic nodal (9+0), and sperm (9+2) axonemes - the constant among these examples are the dynein arm structures. We must also consider the “discordant” ciliary phenotype stemming from *CFAP45* and *CFAP52* mutations in which affected individuals have a mild respiratory phenotype (not supportive for a PCD diagnosis) but pronounced effect on nodal cilia (*situs inversus totalis*) and in the case of *CFAP45*, sperm dyskinesia (consistent with asthenospermia as defined by WHO parameters) in both human and mouse.

Overall, although ENK and IQCD are not disrupted in CFAP45-deficient respiratory cilia and sperm flagella, we cannot rule out that CFAP45-dependent signaling through ENK and IQCD is in some way perturbed, and we acknowledge that this perturbation may contribute to the partial rescue of CFAP45-deficient sperm in response to AMP and ADP (Fig. 7). However, because of reasons cited above including the observation that both CFAP45 and CFAP52 are abnormal or undetectable in several genetically distinct examples of *DNAH11*-mutant cilia, we favor a mechanism that considers this effect to be associated with dynein ATPases at the A-tubule.

We have modified the text on p 18 accordingly:

“CFAP52 and ENKUR (data not shown) as well as AK8 are still detectable in sperm flagella of individual OP-28 II1 and *Cfap45*^{-/-} males (Supplementary Fig. 10h-k), suggesting that CFAP45 deficiency does not disrupt their axonemal localization. However, we cannot rule out that CFAP45-dependent signaling through these or other associations (e.g., IQCD) is perturbed, which may account for the partial rescue of MT sliding in response to AMP and ADP (Fig. 7h).

Also, it is not shown if there are differences for the role of ADP for the inner and outer dynein arms.

d) We address this in more detail in response (g) below.

The data supporting the association of the DNAH11 to the mutant is not as specific or strong as that interactions for CFAP45-52.

e) We have addressed this concern in response (b).

It is provocative that both outer and inner dynein arm proteins are in pull downs with CFAP52. Reference 11 and 12 demonstrate a role for ADP binding to inner dynein arm dynein a. Are their biochemical data related to IDA dyneins?

f) We wish to clarify that CFAP45 (not CFAP52) identified associations with IDA-associated dyneins. We do find these associations intriguing but did not pursue further experiments because both CFAP45 and CFAP52 (as well as DNAH11) show normal axonemal localization in several examples where the IDA structure is missing (determined by TEM as well as IFM using antibodies specific for the ODA component DNAH5 and IDA light chain component DNALI1) in either cytoplasmic dynein arm preassembly mutants such as DNAAF1 or ruler mutants such as CCDC39 and CCDC40 (Fig.4 and Supplementary Fig.6).

Is there NsitePred analysis with proteins at the inner and outer dynein arms?

g) We kindly refer you to Supplementary Fig. 14 that shows the binding probability profile of ATP versus ADP in the four ATPase domains of all fifteen annotated human dynein ATPases. This analysis indicates a trend for inner arm dynein ATPases (DNAH1, DNAH3, DNAH6, DNAH7, DNAH12) to have a higher probability for ADP binding in the AAA1 domain; this suggests that ATP hydrolysis may be differentially regulated (especially at the main AAA1 site) with respect to outer arm dynein ATPases such as DNAH5, DNAH9, and DNAH11. However, a consistent trend is the observation among almost all dynein ATPases that the AAA4 regulatory domain shows a competitive profile for both ATP and ADP, suggesting it is sensitive to fluctuations in levels of either nucleotide. We have modified the text to suggest that CFAP45 mediates adenine nucleotide homeostasis in part through AMP binding, and propose ADP homeostasis via its interaction with AK8 as a potential model that requires additional testing (please see response (d) and (e) to Reviewer #2). Consistent with this latter point, we speculate that loss of CFAP45 disrupts ADP homeostasis, which could affect dynein ATPase / MTD dynamics and consequently a reduction in dynein ATPase force reduction (Supplementary Fig. 15). While NSitePred was valuable in guiding our hypothesis that CFAP45 and CFAP52 could bind nucleotide, our overall conclusion that CFAP45 binds AMP is supported by multiple and complementary approaches (please also see our response (g) to Reviewer #2).

Overall, we have modified the text to reflect our answers including a significantly revised paragraph in the Discussion:

“Our data suggest a functional association between CFAP45 and dynein ATPases including ODA-associated DNAH11 along the A-tubule of axonemal MTDs. Recently, Owa et al. have characterized the Chlamydomonas orthologs to CFAP45 and CFAP52 as luminal proteins of

the B-tubule, although they also identify a reduction of IDA-associated *dynein b* along the A-tubule in respective mutants⁵⁵. Under our experimental conditions using IFM (see **Methods**), we consider the possibility that a) the epitopes of luminal CFAP45 and CFAP52 are accessible without using non-disruptive (i.e., sarkosyl) reagents and / or b) that distinct isoforms of mammalian CFAP45 and CFAP52 have adapted specific functions for the A- and B-tubules of axonemal MTDs. There are notable structural and molecular differences between *Chlamydomonas* and mammalian axonemes, including the reduction of a three-headed ODA (α , β , and γ heavy chain genes) in *Chlamydomonas* to a two-headed ODA (orthologs of β and γ heavy chain genes) in mammals⁵⁶. Furthermore, whereas the α heavy chain gene has no clear ortholog in higher eukaryotes, the β heavy chain gene has duplicated not once but twice⁵⁷ to reflect tissue- and axoneme-specific functions for DNAH11 and DNAH9 in the proximal and distal ciliary regions of mammalian respiratory cilia^{33,47,48} as well as DNAH17 in mammalian sperm flagella (**Supplementary Fig. 12**). A recent study has identified *DNAH17* mutations in males with asthenospermia and characterized DNAH17 as the functional paralog of ODAs in human spermatozoa⁵⁸. Interestingly, we also identified CFAP45 associations to IDA-associated DNAH7 and DNAH10 in porcine respiratory lysates (**Supplementary Table 3** and **Supplementary Fig. 13**). Future studies such as proteomic and cryo-EM analysis of CFAP45- and CFAP52-deficient mammalian cilia and sperm may be helpful in clarifying their functional roles at the A- and B-tubule of axonemal MTDs.”

Reviewer #2

The most significant finding of this manuscript is the identification of CFAP45 as a motile ciliopathy gene.

- a) We appreciate this acknowledgment and have now identified a third unrelated individual with homozygous loss-of-function *CFAP45* mutations (please see below new **Supplementary Fig. 1**). We have modified the title to reflect the novelty and significance of *CFAP45* deficiency as a motile ciliopathy disorder in the human population.

The authors further suggest that CFAP45, CFAP52 and AK8 forms a functional module for ADP homeostasis and indirectly regulates dynein ATPases. Although the importance of ATP/ADP in cilia motility is known, this study provides a specific function for CFAP45. The model put forth by the authors is consistent with the localization of all three proteins to the axoneme and co-IP between them. More direct evidence is provided by the ability of CFAP45 to bind AMP and that microtubule sliding of cfap mutant sperms can be rescued by exogenously added ATP and ADP (Figure 7 and associated supplementary materials). However, this also is an area that needs to be strengthened if the main conclusion is that CFAP45 regulates ADP homeostasis through AK8. Moreover this set of results can be better presented. The statement “CFAP45 immunoprecipitates recovered after high salt lysis of human sperm samples showed affinity to AMP...” is puzzling. Does it mean that CFAP45 was pulled down by AMP beads? Or CFAP is IPed, eluded then pulled down by AMP beads?

- b) We apologize for this lack of clarity –CFAP45 was immunoprecipitated from human sperm using high salt (500mM NaCl) lysis buffer to obtain highly enriched native CFAP45; this

sample was then incubated with AMP agarose (Fig. 7b) as a comparison to the purified recombinant CFAP45 (Fig. 7c, d) that was incubated with AMP agarose. We have rephrased the sentence as follows:

“CFAP45 immunoprecipitates recovered from human sperm (see **Methods**) showed affinity to AMP, particularly 6AH-AMP-agarose.”

We acknowledge your point that additional tests are needed to strengthen the main conclusion that CFAP45 regulates ADP homeostasis through AK8 and have made modifications that we describe in more detail below.

The results with recombinant CFAP45 and AK8 in fig 7c-e potentially suggest that CFAP45 binds to AMP directly. However, the possibility of non-specific binding to beads needs to be ruled out.

- c) We have minimized the possibility of non-specific binding to agarose in two ways: a) samples are precleared with simple agarose beads (stated in the original manuscript, please see *AMP binding assay* in **Methods** section), and b) equal amounts of CFAP45 sample (by concentration) are incubated with AMP conjugated to agarose in four different orientations (equal amounts by packed volume), which serves as an internal control. As shown in Fig. 7b and 7d, CFAP45 consistently shows differential affinity to the four AMP orientations, with strongest affinity to 6AH-AMP-agarose and little to no appreciable affinity to α AH-AMP-agarose. We are cautious to not over-interpret the data but we find it interesting that recombinant AK8 trends in the opposite direction, with strongest affinity to α AH-AMP-agarose. To use an analogy, two runners cannot grab a baton in the same place at the same time; the coordinated transfer requires orientation of the free side towards to awaiting runner. Our structural modeling (Fig. 7i) suggests that the phosphate group of AMP (which is relatively unencumbered in α AH-AMP-agarose) is directed toward AK8 at the proposed binding interface.

More importantly, the activity and properties of AK8 has been described in *Biochem. J.* (2011) 433, 527–534. The authors seem to be at a position to perform biochemical experiments with recombinant proteins to directly ask whether CFAP45 affect the activity of AK8. Results potentially can provide direct and convincing evidence for their model. It will also be interesting to test whether CFAP45 show binding specificity toward AMP and ADP, because an alternative model is that AMP or ADP binding induces a conformational change in CFAP45 that is unrelated to enzymatic conversion by AK.

- d) We appreciate this comment as we have not considered the possibility that AMP (or other nucleotide) binding may modulate CFAP45 activity independently of its interaction with AK8. Based on your comments regarding enzymatic conversion by AK, it will be also useful to test associations with catalytic-null / nucleotide-binding null forms of AK8. We have therefore modified the text to suggest that CFAP45 participates in ciliary adenine nucleotide homeostasis in part through AMP binding. We now reference ADP homeostasis as a potential model that requires additional testing (Supplementary Fig. 15) as outlined above and below.

I found fig. 7g and h difficult to follow. The authors state “Administration of 1mM ATP alone or 1mM ADP alone did not significantly reactivate MT sliding of Cfap45^{-/-} sperm compared to heterozygous control (Fig. 7g); additionally, 1mM AMP alone showed negligible (< 1%) reactivation of MT sliding in control or Cfap45^{-/-} sperm, similar to untreated control (Fig. 7g).” However, it looks to me that 1mM ATP or ADP alone did significantly reactivate MT sliding of Cfap45^{-/-} sperm--from almost none to more than 40%--although not to the same degree as heterozygous control. Similarly, 1mM AMP did not reactivate microtubule sliding in either mutant or control samples (fig. 7g). The key result seems to be that 1mM ATP with 4mM AMP or ADP rescues better than 1mM AMP or ADP in fig 7h. However, the difference is quite modest. In addition, the impact of 4mM ADP or AMP alone needs to be tested to reach a meaningful conclusion. Moreover, if the function of CFAP45 is to provide a platform for forward AK reaction toward ADP production, one would expect a differential response of cfap45 mutant samples to ATP in combination with ADP vs AMP (fig 7h, compare the first two groups of columns).

- e) We agree that 4mM ADP or 4mM ATP alone are important controls and per the observations of Vadnais et al., 2014 (PMID: 24740601), believe including inhibitors to AMP-activated protein kinase as well as adenylate kinase would strengthen the conclusion that CFAP45 promotes ADP homeostasis via the AK pathway. The difference in sliding percentage between 1mM AMP / ADP and 4mM AMP / ADP is significant however, and the seemingly differential response between 4mM AMP and 4mM ADP (first two columns, Fig. 7h) should be placed in the context of the assay conditions. The incubation is 10 minutes long and does not use creatine kinase to replenish ATP, therefore the kinetics of the adenylate kinase reaction (ATP + AMP to ADP and vice versa) can equilibrate with ATP hydrolysis by dyneins within this time span. We cannot complete new experiments in mice within a reasonable timeframe (we estimate at least 6 months). Therefore, we have modified the text to state that AMP and ADP partially rescue microtubule sliding and do not make the distinction that the forward reaction toward ADP is more robust than AMP + ATP in this rescue.

Separate but related, since Cfap45^{-/-} sperm can still beat but show abnormal curvature and amplitude, the impact of adenine nucleotides on mutant sperm beating pattern will be interesting to test.

- f) We agree that testing adenine nucleotides on CFAP45-mutant human sperm will be interesting and important to corroborate partial rescue of MT sliding in CFAP45-mutant mouse sperm in response to AMP and ADP. However, we have further characterized the defect in CFAP45-mutant human sperm (OP-28 II1) by tracking swimming trajectories in high viscosity (1% methyl cellulose) media. The circular swimming phenotype of OP-28 II1 sperm is corrected in high viscosity media but the swimming speed is slower than healthy control sperm by approximately 45% (please see new **Supplementary Fig. 2**). This suggests that external load vis-à-vis high viscosity can correct the waveform but that its force production is still weaker.

We have made several changes to the text to reflect these points including this paragraph in the Discussion:

“Our data also suggest that CFAP45 mediates adenine nucleotide homeostasis, in part through AMP binding and its association with AK8. Additional studies are required to determine if AMP (or other nucleotides) affects the function of CFAP45 independent of its association with AK8, as well as influencing kinetic parameters of the adenylate kinase reaction. The partially restorative effects of AMP and ADP on MT sliding of CFAP45-deficient sperm as well as modelling of the CFAP45*AMP*AK8 interface lead us to speculate that CFAP45 may act as an AMP donor to AK8 and promote the forward reaction toward ADP production. In turn, this potentially explains a mechanism for dynein ATPase regulation, albeit indirectly.”

The final and important piece of evidence is provided by structural modeling. I do not consider myself qualified to evaluate this approach. However, it will be helpful to provide some independent assessment. For example, if the motile cilium proteome is run through NSitePred, how many hits would the program pick up.

- g) As detailed in Chen et al., 2012 (PMID: 22130595), the NsitePred server offers prediction of nucleotide-binding probability. However, NsitePred does not adequately address the collective effect of multiple residue variations between the query sequence and template sequences with known nucleotide-binding sites. This is because the 3D structure of the query sequence is not taken into account in the NsitePred prediction. Given this limitation, we expect that NsitePred would miss some sequences with known nucleotide-binding sites, for example components of the motile cilium proteome. To address the limitation of NsitePred, we actually predicted the 3D structure of the CFAP45*AMP*AK8 complex using advanced simulation techniques to determine the AMP binding site independently from the NsitePred prediction. The extensive isothermal-isobaric simulations (18.96 microseconds) of the large CFAP45*AMP*AK8 complex we describe here represent a significant improvement in refining protein structure compared to techniques available several years ago. As described in our manuscript, the AMP binding site independently predicted using the NsitePred server is in agreement with the simulation-determined AMP binding site. In turn, this is validated by

evidence of CFAP45 binding AMP *in vitro* and the partial rescue of CFAP45-deficient sperm to AMP + ATP.

Regarding the title, “harmonize” at first glance hints at coordination between cilia, which does not seem to be the topic of this manuscript.

- h) As mentioned in response (a), please note the modification of our title to: CFAP45 deficiency causes situs abnormalities and asthenospermia by disrupting an axonemal adenine nucleotide homeostasis module.

Supplementary table 3 is missing.

I also think the list of tested proteins in the 2 hybrid assay should be provided in supplementary materials, instead of “available upon request”. Negative results will be useful to the community and provide a context for the positive results presented in this manuscript.

- i) We apologize for this oversight and have now included Supplementary Table 3 for your review. We have also included the list of 175 genes that were tested by Y2H as new **Supplementary Table 4.**

REVIEWERS' COMMENTS

Reviewer #1 (Remarks to the Author):

The studies provide an putative mechanism for two novel human motile ciliopathies that have been previously not described. The authors have been responsive to the reviews. The authors have further supported their hypothesis with new data and explanations of findings that are logical and acceptable to this reviewer. Original conclusions presented in this paper related to the impact of human mutations on the control of dynein function and ciliary activity will be of interest to the community. The authors fairly point out gaps in our knowledge related to known and possible differences between *Chlamydomonas* and human cilia. These differences will be areas for future investigation that require difficult high resolution studies. Publication of the CFAP45/52/AK8 complex of an adenine nucleotide regulatory machinery will spur future studies that will be important for advancing the field.

Reviewer #2 (Remarks to the Author):

The authors have sufficiently addressed my questions. I have no further comments.

Reviewer #3 (Remarks to the Author):

I only have some comments on technical issues of the MD simulations.

1. What did they mean by saying "manually dock..."? Did they use any software?
2. As I know, the force field parameters of AMP is available in Amber. Why did the authors obtain the charges using the ab initio calculation?
3. Why did the authors set the temperature to 340K in the MD simulations. It is significantly higher than the room temperature.

Dear Dr. David,

On behalf of the authors, we would like to thank you again for evaluating manuscript NCOMMS-19-36614A, and we are pleased to receive news that you and the editorial team deem our work suitable for publication in Nature Communications.

Please see our responses (blue font) to Reviewer's Comments below; our responses to Reviewer #3 have been added to the Methods section under the heading **Theoretical model of the CFAP45•AMP•AK8 complex** as both "clean" and "marked-up" manuscript files.

REVIEWERS' COMMENTS

Reviewer #1 (Remarks to the Author):

The studies provide an putative mechanism for two novel human motile ciliopathies that have been previously not described. The authors have been responsive to the reviews. The authors have further supported their hypothesis with new data and explanations of findings that are logical and acceptable to this reviewer. Original conclusions presented in this paper related to the impact of human mutations on the control of dynein function and ciliary activity will be of interest to the community. The authors fairly point out gaps in our knowledge related to known and possible differences between Chlamydomonas and human cilia. These differences will be areas for future investigation that require difficult high resolution studies. Publication of the CFAP45/52/AK8 complex of an adenine nucleotide regulatory machinery will spur future studies that will be important for advancing the field.

We thank Reviewer #1 for their extensive time and effort in reviewing the manuscript and raising critical points that guided new experiments and ultimately improved the breadth and interpretation of the datasets. We appreciate the concise summary and recognition that our study will facilitate future studies relating human ciliopathy genes to dynein function and ciliary activity.

Reviewer #2 (Remarks to the Author):

The authors have sufficiently addressed my questions. I have no further comments.

We also thank Reviewer #2 for their extensive time and effort in reviewing the manuscript and raising critical points that ultimately sharpened our overall conclusion of the datasets.

Reviewer #3 (Remarks to the Author):

I only have some comments on technical issues of the MD simulations.

1. What did they mean by saying "manually dock..."? Did they use any software?

We thank Reviewer #3 for critically reviewing our manuscript. We have provided the software information for the manual docking and in the Methods section under the heading Theoretical model of the CFAP45•AMP•AK8 complex, we replaced the previous statement,

“The Trichoplein domain was then manually docked onto the KD2 domain to form the CFAP45•AK8 dimer. AMP was then manually docked into a cavity at the interface of the dimer in such a way that AMP was primarily interacting with CFAP45.”

with the revised statement,

“Using MacPyMOL (PyMOL v1.7.0.3 Enhanced for Mac OS X), we manually placed the Trichoplein domain in the vicinity of the KD2 domain to form a crude CFAP45•AK8 dimer, and then manually inserted AMP into a cavity at the interface of the dimer in such a way that AMP was primarily interacting with CFAP45.”

2. As I know, the force field parameters of AMP is available in Amber. Why did the authors obtain the charges using the ab initio calculation?

To obtain more realistic atomic charges at C1' and C4' of the AMP molecule, we derived the RESP charges of AMP from our ab initio calculation of the intact AMP molecule. This was because the RESP charges of AMP available at the RESP ESP charge DDataBase (<https://upjv.q4md-forcefieldtools.org/REDDB/projects/F-90/>) were derived from splicing together two fragments from adenosine and methyl phosphate.

3. Why did the authors set the temperature to 340K in the MD simulations. It is significantly higher than the room temperature.

We performed the MD simulations using FF12MC at 340 K instead of 298 K for better conformational sampling, according to the reports that the protein folding times derived from the MD simulations employing FF12MC are inversely related to the simulation temperatures and in excellent agreement with the experimentally determined folding times at different temperatures (references already included in previous submission, Pang, 2016⁵² and Pang, 2017⁵³).